# Efficient Algorithms for Recognizing Weighted Tree-Adjoining Languages

**Alexandra Butoi**[1]    **Tim Vieira**[1]    **Ryan Cotterell**[1]    **David Chiang**[2]

[1]ETH Zürich    [2]University of Notre Dame
{alexandra.butoi, ryan.cotterell}@inf.ethz.ch
dchiang@nd.edu    tim.f.vieira@gmail.com

## Abstract

The class of tree-adjoining languages can be characterized by various two-level formalisms, consisting of a context-free grammar (CFG) or pushdown automaton (PDA) controlling another CFG or PDA. These four formalisms are equivalent to tree-adjoining grammars (TAG), linear indexed grammars (LIG), pushdown-adjoining automata (PAA), and embedded pushdown automata (EPDA). We define semiring-weighted versions of the above two-level formalisms, and we design new algorithms for computing their stringsums (the weight of all derivations of a string) and allsums (the weight of all derivations). From these, we also immediately obtain stringsum and allsum algorithms for TAG, LIG, PAA, and EPDA. For LIG, our algorithm is more time-efficient by a factor of $\mathcal{O}(n|\mathcal{N}|)$ (where $n$ is the string length and $|\mathcal{N}|$ is the size of the nonterminal set) and more space-efficient by a factor of $\mathcal{O}(|\Gamma|)$ (where $\Gamma$ is the size of the stack alphabet) than the algorithm of Vijay-Shanker and Weir (1989). For EPDA, our algorithm is both more space-efficient and time-efficient than the algorithm of Alonso et al. (2001) by factors of $\mathcal{O}(|\Gamma|^2)$ and $\mathcal{O}(|\Gamma|^3)$, respectively. Finally, we give the first PAA stringsum and allsum algorithms.

## 1 Introduction

Weir (1992) introduced a hierarchy of formal languages whose first level ($\mathcal{L}_1$) is the class of context-free languages and second level ($\mathcal{L}_2$) is the class of tree-adjoining languages. Just as context-free languages can be characterized by both context-free grammars and pushdown automata, tree-adjoining languages are characterized by multiple formalisms as well, including tree-adjoining grammars (TAG; Joshi et al., 1975), linear indexed grammars (LIG; Gazdar, 1988), embedded pushdown automata (EPDA; Vijay-Shanker, 1987), and pushdown-adjoining automata (PAA; Butoi et al., 2023).

Tree-adjoining languages can further be characterized through the mechanism of *control* (Weir, 1992), which yields various *two-level* formalisms. Specifically, we have shown that

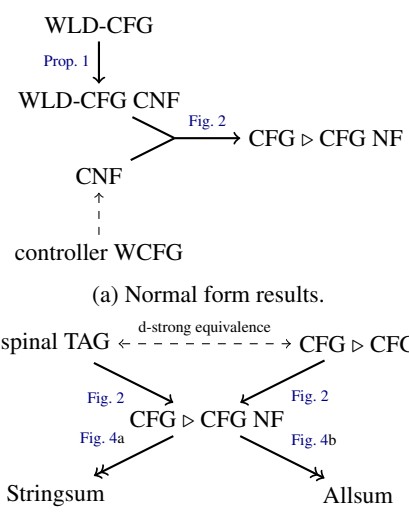

(a) Normal form results.

(b) Stringsum and allsum results.

Figure 1: Overview of the results presented in this paper. Solid lines are new results shown in the paper; dashed lines are old results. An arrow $X \to Y$ means "$X$ can be converted to $Y$"; an arrow $X \twoheadrightarrow Y$ means "$X$ has an algorithm for computing $Y$". We only show the conversions and algorithms for CFG ▷ CFG; the results for the other three two-level formalisms are analogous.

CFGs controlled by CFGs, CFGs controlled by PDAs, PDAs controlled by CFGs, and PDAs controlled by PDAs are equivalent to TAG, LIG, PAA, and EPDA, respectively, in a strict sense called *d-strong* equivalence (Butoi et al., 2023).

When designing statistical parsers for tree-adjoining formalisms, it is often useful to consider their semiring-weighted generalizations. In this paper, we introduce semiring-weighted versions of the above two-level formalisms, and we give new, more efficient algorithms for computing stringsums (the total weight of all derivations of a string) in these formalisms.

Lang (1974) gives a recognition algorithm for (what we call) simple PDAs, which can pop and push at most one stack symbol. We have shown that this algorithm is suboptimal, and that (what we call) top-down PDAs, which always pop exactly one stack symbol, allow computing stringsums more

efficiently (Butoi et al., 2022).

Existing definitions of LIG (Vijay-Shanker and Weir, 1989; Joshi et al., 1991; Kanazawa, 2014) are equivalent to CFGs controlled by simple PDAs, and existing recognition algorithms (e.g., Vijay-Shanker and Weir, 1989) have the same limitation that makes Lang's algorithm suboptimal. By using a top-down PDA as a controller instead, we obtain a new stringsum algorithm that is more space-efficient and runs asymptotically faster. Additionally, we obtain an algorithm for allsums (the total weight of all derivations) that is more space-efficient.

All stringsum algorithms that operate directly on EPDAs that we are aware of (Alonso et al., 2000, 2001) are designed for specific types of EPDAs, failing to take advantage of the structure of computation in top-down PDAs. We design a new stringsum algorithm for a subclass of EPDA that is equivalent to a top-down PDA controlled by a top-down PDA, and we obtain both time and space improvements over these previous algorithms. Additionally, we design a more space-efficient EPDA allsum algorithm.

Our algorithms assume that CFGs are given in Chomsky normal form, and for PDAs, we define a new normal form that is exactly analogous to the Chomsky normal form. We show that applying these normal forms to controllers and controllees induces, for free, normal forms for the two-level formalisms and for TAG, LIG, PAA, and EPDA. The conversions into the normal forms for all these formalisms become simpler than direct conversions, as they only require CFG or PDA conversions. We leave extensions of our stringsum and allsum algorithms to general CFGs/PDAs for future work.

The main contributions of this paper are:

- Semiring-weighted versions of the formalisms CFG ▷ CFG (§3), PDA ▷ CFG, CFG ▷ PDA, and PDA ▷ PDA (App. B.1).
- Normal forms for these formalisms that arise, for free, from the normal forms of the controller/-controllee CFGs/PDAs (§3.2 and App. B.2).
- Stringsum algorithms for CFG ▷ CFG (§4), PDA ▷ CFG, and PDA ▷ PDA (App. C.2) that are more time-efficient or space-efficient than the existing algorithms for their equivalent LIG and EPDA.
- The first stringsum algorithm for CFG ▷ PDA, and therefore, for PAA (App. C.2).
- Algorithms for computing allsums in these two-level formalisms (§5 and App. D).

## 2 Preliminaries

Let $[i{:}j]$ denote the sequence of integers $(i, \ldots, j)$. If $s$ is a string, we write $|s|$ for the length of $s$, $s_i$ for the $i^{\text{th}}$ symbol of $s$, and $s_{(i:j)}$ for the substring $s_{i+1} \cdots s_j$.

### 2.1 Semiring-Weighted Languages

Throughout this paper, we assume that $\mathcal{W} = (W, \oplus, \otimes, \mathbf{0}, \mathbf{1})$ is a commutative semiring. We also sometimes assume that $\mathcal{W}$ is $\omega$-continuous, which makes it possible to take (countably) infinite sums. Please see App. A.1 for definitions of these terms. Readers not familiar with semirings may safely assume that $W = \mathbb{R}_{\geq 0} \cup \{\infty\}$, $\mathbf{0} = 0$, $\mathbf{1} = 1$, and $\oplus$ and $\otimes$ are the usual addition and multiplication operations, respectively.

**Definition 1.** *An **alphabet** $\Sigma$ is a non-empty finite set of **symbols**. A **language** over $\Sigma$ is a subset of $\Sigma$'s **Kleene closure** $\Sigma^* \overset{\text{def}}{=} \bigcup_{n \geq 0} \Sigma^n$. A **weighted language** over $\Sigma$ with weights from $\mathcal{W} = (W, \oplus, \otimes, \mathbf{0}, \mathbf{1})$ is a mapping from $\Sigma^*$ to $W$.*

This paper considers *weighted* formalisms, where the weights are taken from $\mathcal{W}$. Rather than just producing or recognizing strings from a formal language, these formalisms define a weighting function over $\Sigma^*$. When $\mathcal{W}$ is the boolean semiring $(\{0, 1\}, \vee, \wedge, 0, 1)$, we recover the usual notion of a formal language defined as a set of strings.

### 2.2 Weighted Context-Free Grammars

**Definition 2.** *A **weighted context-free grammar** (WCFG) over a semiring $\mathcal{W} = (W, \oplus, \otimes, \mathbf{0}, \mathbf{1})$ is a tuple $\mathbf{G} = (\mathcal{N}, \Sigma, \mathcal{R}, \mathrm{w}, S)$, where $\mathcal{N}, \Sigma$ are finite sets of nonterminal symbols and terminal symbols, respectively, $\mathcal{R} \subseteq \mathcal{N} \times (\Sigma \cup \mathcal{N})^*$ is a finite set of productions, $\mathrm{w} \colon \mathcal{R} \to W$ is a production-weighting function, and $S \in \mathcal{N}$ is the start symbol.*

*We write a production $(X, \boldsymbol{\alpha}) \in \mathcal{R}$ as $X \to \boldsymbol{\alpha}$, and if $\mathrm{w}(X \to \boldsymbol{\alpha}) = w$, we write $X \xrightarrow{w} \boldsymbol{\alpha}$.*

A WCFG produces strings by starting from the symbol $S$ and repeatedly replacing the leftmost nonterminal $X$ with the right-hand side of a production $X \to \boldsymbol{\alpha}$ until no more nonterminals are left. In the following definitions, let $\mathbf{G} = (\mathcal{N}, \Sigma, \mathcal{R}, \mathrm{w}, S)$ be a WCFG.

**Definition 3.** *If $\boldsymbol{\alpha} = \boldsymbol{\beta}_1 X \boldsymbol{\beta}_2$ and $\boldsymbol{\alpha}' = \boldsymbol{\beta}_1 \boldsymbol{\gamma} \boldsymbol{\beta}_2$ are sequences of terminals and nonterminals of $\mathbf{G}$, where $\boldsymbol{\beta}_1 \in \Sigma^*$ and $\boldsymbol{\beta}_2 \in (\Sigma \cup \mathcal{N})^*$, and $p =$*

$(X \xrightarrow{w} \boldsymbol{\gamma})$ is a production of $\mathbf{G}$, we write $\boldsymbol{\alpha} \xRightarrow{p} \boldsymbol{\alpha}'$ to denote that $\boldsymbol{\alpha}$ *derives* $\boldsymbol{\alpha}'$ in one step using production $p$.

**Definition 4.** *A **partial derivation** in $\mathbf{G}$ from $\boldsymbol{\alpha}_0$ to $\boldsymbol{\alpha}_n$ is a sequence of steps $d = \boldsymbol{\alpha}_0 \xRightarrow{p_1} \cdots \xRightarrow{p_n} \boldsymbol{\alpha}_n$. We write $\boldsymbol{\alpha}_0 \xRightarrow{*} \boldsymbol{\alpha}_n$ to assert that some partial derivation from $\boldsymbol{\alpha}_0$ to $\boldsymbol{\alpha}_n$ exists, or to denote the set of all such partial derivations. If $\boldsymbol{\alpha}_0 = S$ and $\boldsymbol{\alpha}_n = \boldsymbol{s} \in \Sigma^*$, we call $d$ a **derivation** of $\boldsymbol{s}$, or that $\mathbf{G}$ derives $\boldsymbol{s}$.*

*The **weight** of $d$ is the product of its production weights,*

$$\mathbf{w}(d) \overset{\text{def}}{=} \bigotimes_{i=1}^n \mathrm{w}(p_i).$$

*We denote by $\mathcal{D}(\mathbf{G}, \boldsymbol{s})$ the set of all derivations in $\mathbf{G}$ of $\boldsymbol{s}$ and by $\mathcal{D}(\mathbf{G})$ the set of all derivations in $\mathbf{G}$.*

**Definition 5.** *The **stringsum** $\mathbf{w}(\mathbf{G}, \boldsymbol{s})$ of a string $\boldsymbol{s}$ under $\mathbf{G}$ is the total weight of all derivations in $\mathbf{G}$ for $\boldsymbol{s}$,*

$$\mathbf{w}(\mathbf{G}, \boldsymbol{s}) \overset{\text{def}}{=} \bigoplus_{d \in \mathcal{D}(\mathbf{G}, \boldsymbol{s})} \mathbf{w}(d).$$

**Definition 6.** *The **allsum** $\mathbf{w}(\mathbf{G})$ of $\mathbf{G}$ is the total weight of all its derivations,*

$$\mathbf{w}(\mathbf{G}) \overset{\text{def}}{=} \bigoplus_{d \in \mathcal{D}(\mathbf{G})} \mathbf{w}(d).$$

## 3 Semiring-Weighted CFG ▷ CFG

Weir (1992) defined a hierarchy of formal languages and showed that its second level, which is commonly known as the class of tree-adjoining languages, can be obtained through the mechanism of control, using a CFG (the controllee) whose derivations are controlled by another CFG (the controller). In previous work (Butoi et al., 2023), we extended this mechanism to PDAs, both as controllers and controllees, and obtained four distinct formalisms by mixing a controller CFG or PDA with a controllee CFG or PDA. In this paper, we use *semiring-weighted* versions of these formalisms. We give a formal definition of a weighted CFG ▷ CFG here (see App. B.1 for the other three definitions).

We denote such a grammar by $\mathbf{G}_1 \triangleright \mathbf{G}_2$, where $\mathbf{G}_1$ is the controller and $\mathbf{G}_2$ is the controllee. Additionally, we use symbols $X, Y, Z, \ldots$ and $a, b, c, \ldots$ for nonterminals and terminals of the controllee, and $A, B, C, \ldots$ and $a, b, c, \ldots$ for nonterminals and terminals of the controller.

### 3.1 Definition

A weighted labeled distinguished CFG (WLD-CFG) is simply a WCFG where each production has a label, and its right-hand side has one "distinguished" occurrence of a nonterminal symbol. These two extensions allow its derivations to be controlled by another formalism.

**Definition 7.** *A **weighted labeled distinguished context-free grammar** (WLD-CFG) over a semiring $\mathcal{W} = (W, \oplus, \otimes, \mathbf{0}, \mathbf{1})$ is a tuple $\mathbf{G} = (\mathcal{N}, \Sigma, L, \mathcal{R}, \mathrm{w}, S)$, where*

- *$\mathcal{N}$, $\Sigma$, and $L$ are finite sets of nonterminal symbols, terminal symbols, and labels, respectively,*

- *$\mathcal{R} \subseteq L \times \mathbb{N} \times \mathcal{N} \times (\mathcal{N} \cup \Sigma)^*$ is a finite set of productions,*

- *$\mathrm{w} \colon \mathcal{R} \to W$ is a production-weighting function, and*

- *$S \in \mathcal{N}$ is the start symbol.*

*If $(\ell, i, A, \beta_1 \cdots \beta_n)$ is a production in $\mathcal{R}$, we must have either $i = 0$, which we write as*

$$\ell : A \to \beta_1 \cdots \beta_n$$

*or $1 \leq i \leq n$ and $\beta_i \in \mathcal{N}$, which we write as*

$$\ell : A \to \beta_1 \cdots \beta_{i-1} \overset{\smile}{\beta_i} \beta_{i+1} \cdots \beta_n.$$

Weir (1992) gave two definitions of a derivation in an LD-CFG (the controllee) controlled by another CFG (the controller). In his first definition, the controllee nonterminals keep a record of the productions used, called a **control word** (a string in $L^*$). At the end of the derivation, each control word is checked for membership in the language of the controller. In his second definition, the controllee nonterminals run a controller derivation. When the controller generates a label $\ell \in L$, it causes the controllee to apply production $\ell$. We use the latter definition, which allows one to think of the controller and the controllee as a single grammar, merging their productions into a single set of rules.

For any sets $\mathcal{X}$ and $\mathcal{Y}$, we define $\mathcal{X}[\mathcal{Y}] = \{X[Y] \mid X \in \mathcal{X}, Y \in \mathcal{Y}\}$, where $X[Y]$ is just another way of writing the ordered pair of $X$ and $Y$. If $\boldsymbol{\alpha} = X_1 \cdots X_k \in \mathcal{X}^*$ is a sequence of nonterminals, we use the notation $\boldsymbol{\alpha}[Y] = X_1[Y] \cdots X_k[Y]$ for any $Y \in \mathcal{Y}$.

To make the following definition more readable, we assume that the controller's right-hand sides are either a single label or a string of nonterminals, and that the controllee right-hand sides are either a single terminal or a string of nonterminals with exactly one distinguished position. It would be possible, but more tedious, to write the definition for the general case.

**Definition 8.** *Let $\mathbf{G}_1$ be a WCFG with nonterminals $\mathcal{N}_1$ and terminals $L$, called the **controller**, and let $\mathbf{G}_2$ be a WLD-CFG with nonterminals $\mathcal{N}_2$ and labels $L$, called the **controllee**. Then $\mathbf{G}_1 \triangleright \mathbf{G}_2$ is a rewriting system with the following rules:*

- *If $(A \xrightarrow{w} \beta)$ is a production of $\mathbf{G}_1$ where $\beta \in \mathcal{N}_1^*$, then $\mathbf{G}_1 \triangleright \mathbf{G}_2$ has a rule $X[A \cdot\cdot] \xrightarrow{w} X[\beta \cdot\cdot]$ for each $X \in \mathcal{N}_2$.*

- *If $(A \xrightarrow{w_1} \ell)$ is a production of $\mathbf{G}_1$ and $(\ell \colon X \xrightarrow{w_2} \alpha_1 \breve{Y} \alpha_2)$ is a production of $\mathbf{G}_2$, then $\mathbf{G}_1 \triangleright \mathbf{G}_2$ has a rule $X[A \cdot\cdot] \xrightarrow{w_1 \otimes w_2} \alpha_1[S]\, Y[\cdot\cdot]\, \alpha_2[S]$.*

*A derivation in $\mathbf{G}_1 \triangleright \mathbf{G}_2$ starts with $S[S]$. If there is a rule $p = (X[A \cdot\cdot] \xrightarrow{w} \alpha_1 Y[\beta_1 \cdot\cdot]\alpha_2)$, then for any $\alpha_0 \in \Sigma^*$, $\beta_2 \in \mathcal{N}_2^*$, and $\alpha_3 \in (\Sigma \cup \mathcal{N}_1[\mathcal{N}_2^*])^*$, we write*

$$\alpha_0 X[A\beta_2]\alpha_3 \xRightarrow{p} \alpha_0 \alpha_1 Y[\beta_1\beta_2]\alpha_2\alpha_3.$$

*Similarly, if there is a rule $p = (X[A] \xrightarrow{w} \alpha)$, then for any $\alpha_0 \in \Sigma^*$, and $\alpha_3 \in (\Sigma \cup \mathcal{N}_1[\mathcal{N}_2^*])^*$, we write*

$$\alpha_0 X[A]\alpha_3 \xRightarrow{p} \alpha_0 \alpha \alpha_3.$$

*We write $\overset{*}{\Rightarrow}$ to denote a derivation with zero or more steps, as in Def. 4. If $S[S] \overset{*}{\Rightarrow} s \in \Sigma^*$, we say that $\mathbf{G}_1 \triangleright \mathbf{G}_2$ **derives** $s$.*

**Example 1.** *Consider the following $\mathbf{G}_1 \triangleright \mathbf{G}_2$ that generates the language $\{a^n b^n c^n d^n \mid n \in \mathbb{N}\}$. The controller is a WCFG $\mathbf{G}_1$ with the start symbol $S_1$ and the set of productions*

$$\mathcal{R}_1 = \left\{ \begin{array}{l} S_1 \xrightarrow{1} TL_3 \\ T \xrightarrow{1} L_1 T L_2 \\ T \xrightarrow{1} \varepsilon \\ L_i \xrightarrow{1} \ell_i \quad (i \in [1:3]) \\ S_1 \xrightarrow{1} \ell_i \quad (i \in [4:7]) \end{array} \right\}$$

*and the controllee is a WLD-CFG $\mathbf{G}_2$ with start symbol $S_2$ and the set of productions*

$$\mathcal{R}_2 = \left\{ \begin{array}{ll} \ell_1 \colon S_2 \xrightarrow{1} A\breve{S}_2 D & \ell_4 \colon A \xrightarrow{1} a \\ \ell_2 \colon S_2 \xrightarrow{1} B\breve{S}_2 C & \ell_5 \colon B \xrightarrow{1} b \\ \ell_3 \colon S_2 \xrightarrow{1} \varepsilon & \ell_6 \colon C \xrightarrow{1} c \\ & \ell_7 \colon D \xrightarrow{1} d \end{array} \right\}.$$

*Below is a derivation of the string* aabbccdd.

$$
\begin{aligned}
S_2[S_1] &\Rightarrow S_2[TL_3] \\
&\Rightarrow S_2[L_1 T L_2 L_3] \\
&\Rightarrow A[S_1]S_2[TL_2L_3]D[S_1] \\
&\Rightarrow aS_2[TL_2L_3]D[S_1] \\
&\Rightarrow aS_2[L_1TL_2L_2L_3]D[S_1] \\
&\overset{*}{\Rightarrow} aaS_2[L_2L_2L_3]D[S_1]D[S_1] \\
&\Rightarrow aaB[S_1]S_2[L_2L_3]C[S_1]D[S_1]D[S_1] \\
&\Rightarrow aabS_2[L_2L_3]C[S_1]D[S_1]D[S_1] \\
&\overset{*}{\Rightarrow} aabbccdd.
\end{aligned}
$$

Analogous definitions for when the controller is a WPDA and/or the controllee is a WLD-PDA are given in App. B.1.

### 3.2 Normal Form

In this section, we define a normal form for CFG $\triangleright$ CFG that will help us design fast stringsum and allsum algorithms. Interestingly, this normal form arises naturally from the normal forms of the controllee and the controller. Analogous normal forms are derived for PDA $\triangleright$ CFG, CFG $\triangleright$ PDA, and PDA $\triangleright$ PDA in App. B.2.3.

**Definition 9.** *A WCFG is in **Chomsky normal form** if all of its productions are of one of the following types: (1) $S \to \varepsilon$, (2) $X \to a$, or (3) $X \to YZ$, where $S, X, Y, Z \in \mathcal{N}$ and $a \in \Sigma$. Moreover, $S$ does not appear on the right-hand side of any production.*

*A WLD-CFG is in Chomsky normal form if all of its productions are of type (1) or (2) above, (3a) $X \to \breve{Y} Z$, or (3b) $X \to Y \breve{Z}$.*

**Proposition 1.** *For any WCFG $\mathbf{G}_1$ with weights in an $\omega$-continuous semiring, there is a WCFG in Chomsky normal form that defines the same weighted language as $\mathbf{G}_1$.*

*For any WLD-CFG $\mathbf{G}_2$ with weights in an $\omega$-continuous semiring, there is a WLD-CFG in Chomsky normal form that is equivalent to $\mathbf{G}_2$.*

| $\mathbf{G}_1$ | $\mathbf{G}_2$ | $\mathbf{G}_1 \triangleright \mathbf{G}_2$ | name |
|---|---|---|---|
| $S \xrightarrow{w_1} \ell$ | $\ell : S \xrightarrow{w_2} \varepsilon$ | $S[S] \xrightarrow{w_1 \otimes w_2} \varepsilon$ | (epsilon) |
| $A \xrightarrow{w_1} \ell$ | $\ell : X \xrightarrow{w_2} a$ | $X[A] \xrightarrow{w_1 \otimes w_2} a$ | (terminal) |
| $A \xrightarrow{w_1} \ell$ | $\ell : X \xrightarrow{w_2} \breve{Y} Z$ | $X[A \cdot\cdot] \xrightarrow{w_1 \otimes w_2} Y[\cdot\cdot]Z[S]$ | (left pop) |
| $A \xrightarrow{w_1} \ell$ | $\ell : X \xrightarrow{w_2} Y \breve{Z}$ | $X[A \cdot\cdot] \xrightarrow{w_1 \otimes w_2} Y[S]Z[\cdot\cdot]$ | (right pop) |
| $A \xdashrightarrow{w_1} BC$ | | $X[A \cdot\cdot] \xrightarrow{w_1} X[BC \cdot\cdot]$ | (push) |

Figure 2: Normal form of CFG ▷ CFG. Each row shows a rule from $\mathbf{G}_1$ (in Chomsky normal form) and possibly a rule from $\mathbf{G}_2$ (in Chomsky normal form) and the resulting normal-form rule of $\mathbf{G}_1 \triangleright \mathbf{G}_2$ along with its name.

*Proof.* See App. B.2.1. □

Recall that a CFG ▷ CFG is defined as a LD-CFG whose derivations are controlled by another CFG. When we consider only W(LD-)CFGs in Chomsky normal form, we obtain, for free, a normal form for CFG ▷ CFG only by mixing their rules, as shown in Fig. 2. For the epsilon rule, in principle the left-hand side could be $S[A]$ for any $A$, but such rules would never be used.

Due to d-strong equivalence, this normal form also induces a normal form for TAG. But, in order to convert a TAG into the normal form, one needs to first extract the controller and the controllee as shown by Butoi et al. (2023), then convert these to the Chomsky normal form, merge their rules, and convert them back to a TAG, also shown by Butoi et al. (2023) (see App. E for a complexity analysis of these transformations). Although these transformations add some extra complexity, the conversion into the normal form becomes simpler as it only requires (LD-)CFG conversions, rather than a direct TAG conversion. We leave a direct conversion for future work.

## 4 Computing Stringsums

In this section, we give a deduction system for computing stringsums of a particular string $s$ in CFG ▷ CFG in normal form. Stringsums of a string $s$ can be computed analogously in PDA ▷ CFG, CFG ▷ PDA, and PDA ▷ PDA; we show the deduction systems for these in App. C.2. Due to d-strong equivalence, these deduction systems can also be used for spinal TAG, LIG, EPDA, and spinal PAA by first converting them to their equivalent two-level formalism. Our algorithms have improved space requirements and run asymptotically faster than the best algorithms for LIG and EPDA. Moreover, it is the only stringsum algorithm for PAA

that we are aware of.

### 4.1 Pop Computations

In order to compute stringsums efficiently, we decompose derivations into shareable smaller parts. Our decomposition is based on a concept that has been extensively used in the literature (Kuhlmann et al., 2011; Butoi et al., 2022) for decomposing runs of pushdown automata, namely pop computations. In the context of WPDAs, a pop computation is a partial derivation that has the overall effect of popping a single stack symbol, leaving the rest of the stack unchanged (Butoi et al., 2022). We can define pop computations of CFG ▷ CFG with a similar interpretation: partial derivations that "pop" a single leftmost nonterminal symbol, leaving the rest of the nonterminals unchanged. We define pop computations of CFG ▷ CFG formally below. The pop computations of PDA ▷ CFG, CFG ▷ PDA, and PDA ▷ PDA are defined similarly; we present these in App. C.1.

**Definition 10.** *Let $\mathbf{G}_1$ be a controller WCFG and $\mathbf{G}_2$ be a WLD-CFG, with nonterminal alphabets $\mathcal{N}_1$ and $\mathcal{N}_2$, respectively. Let $s \in \Sigma^*$ be a string of length $|s| = n$. A **pop computation** of $\mathbf{G}_1 \triangleright \mathbf{G}_2$ of*

*type* , *where $0 \le i \le j \le k \le l \le n$, $X, Y \in \mathcal{N}_2$, and $A \in \mathcal{N}_1$, is a pair of partial derivations $(d_1, d_2)$, where*

$$d_1 \in (X[A] \xRightarrow{*} s_{(i:j)} Y[\varepsilon] \Upsilon_2)$$

$$d_2 \in (\Upsilon_2 \xRightarrow{*} s_{(k:l)}).$$

*The weight of the pop computation is $\mathbf{w}(d_1, d_2) = \mathbf{w}(d_1) \otimes \mathbf{w}(d_2)$.*

*A **pop computation** of type*  *is a partial derivation $X[A] \xRightarrow{*} s_{(i:j)}$.*

In words, a pop computation of a CFG ▷ CFG pops a single controller symbol and derives a string possibly with a *gap*. Fig. 3 shows a visual representation of a pop computation of a CFG ▷ CFG.

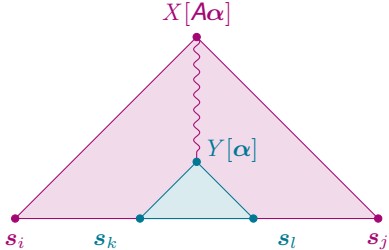

Figure 3: A pop computation of a CFG ▷ CFG is a partial derivation possibly with a gap (the teal part). First, $X[A\alpha]$ derives $s_{(i:k]}$ and $Y[\alpha]$, popping one nonterminal $A$. Then $Y[\alpha]$ derives the gap ($s_{(k:l]}$). Finally, the purple part of the partial derivation resumes, deriving $s_{(l:j]}$. The weight of the pop computation is the weight of the purple part of the partial derivation.

## 4.2 Deduction System

We compute stringsums of CFG ▷ CFG efficiently by exploiting the decomposable structure of the pop computations. We give the stringsum algorithm as a weighted deduction system (Goodman, 1999), shown in Fig. 4a.

**Items.** The deductive system has two types of items, which correspond to the two types of pop computations (one with a gap and one without a gap). The weights of these items are the total weights of the pop computations of those types.

**Deduction rules.** We distinguish several types of pop computations based on their first production and include a deduction rule for each. The weight of each pop computation can be derived recursively from the weights of shorter pop computations. The weight of each deduction rule is the weight of the production in the side condition. Notice that the deduction system only shows how to derive further items, leaving the order in which these items should be derived by an algorithm unspecified.

**Goal item.** The goal item is $\underset{0}{\overset{S[S]}{\triangle}}_n$, which stands for all derivations from $S[S]$ to $s$, where $|s| = n$. Its weight is the total weight of all such derivations, which is exactly the stringsum.

## 4.3 Correctness

We prove the correctness of the deduction system for CFG ▷ CFG using the following theorem.

**Theorem 1.** *Let* $\mathbf{G}_1$ *be a controller WCFG with nonterminal alphabet* $\mathcal{N}_1$ *and start symbol* $S$*, and* $\mathbf{G}_2$ *a WLD-CFG with nonterminal alphabet* $\mathcal{N}_2$*. Let* $s \in \Sigma^*$ *be a string of length* $|s| = n$*.*

(a) *The weight* $w$ *of an item* $\underset{i}{\overset{X[A]}{\triangle}}_j$ *, where* $X \in \mathcal{N}_2$*, and* $0 \le i \le j \le n$*, is the total weight of all pop computations of this type.*

(b) *The weight of an item* $\underset{i}{\overset{X[A\,\cdot\cdot]}{\triangle}}\underset{j\ \ k}{\overset{Y[\cdot\cdot]}{}}_l$ *, where* $A \in \mathcal{N}_1$*,* $X, Y \in \mathcal{N}_2$*, and* $0 \le i \le j \le k \le l \le n$*, is the total weight of all pop computations of this type.*

*Proof.* Define the *span* of item $\underset{i}{\overset{X[A]}{\triangle}}_j$ to be $j - i$, and that of item $\underset{i}{\overset{X[A\,\cdot\cdot]}{\triangle}}\underset{j\ \ k}{\overset{Y[\cdot\cdot]}{}}_l$ to be $j - i + l - k$. We proceed by induction on the span $\ell$.

**Base Case.** When $\ell = 1$, the pop computation consists of a single step $X[A] \overset{p}{\Rightarrow} a$. The weight of item $\underset{i-1}{\overset{X[A]}{\triangle}}_i$ is w($p$) according to inference rule (terminal).

**Inductive Step.** Assume that the statement holds for pop computations with span at most $(\ell - 1)$ and consider pop computations with span $\ell$. The weight of all pop computations of a certain type is the sum of various cases, distinguished by their first production.

**Starting with left pop rules.** Any pop computation starting with $X[A]$ and a left pop production $p$ has the form $(X[A] \overset{p}{\Rightarrow} Y[\varepsilon]Z[S], Z[S] \overset{*}{\Rightarrow} s_{(j:k]})$.

By the inductive hypothesis, the weight $w_1$ of item $\underset{j}{\overset{Z[S]}{\triangle}}_k$ is the total weight of all partial derivations such that $Z[S] \overset{*}{\Rightarrow} s_{(j:k]}$. By distributivity, the total weight of all pop computations of the above form is $\mathbf{w}(p) \otimes w_1$. This weight is added to the weight of the item $\underset{i}{\overset{X[A\,\cdot\cdot]}{\triangle}}\underset{k\ \ l}{\overset{Y[\cdot\cdot]}{}}_o$ in inference rule (left pop).

Since the rules are symmetric, a similar argument can be used for pop computations that start with right pop rules.

**Starting with push rules.** Any pop computation starting with $X[A]$ and a push production $p$ has the

| name | (a) stringsum | (b) allsum | side condition |
|---|---|---|---|

(epsilon) $\qquad$ $n = 0$ $\qquad$ $S[S] \xrightarrow{w} \varepsilon$

(terminal) $\qquad$ $\boldsymbol{s}_i = a$ $\qquad$ $X[A] \xrightarrow{w} a$

(left pop) $\qquad$ $X[A\,\cdot\cdot] \xrightarrow{w} Y[\cdot\cdot]\,Z[S]$

(right pop) $\qquad$ $X[A\,\cdot\cdot] \xrightarrow{w} Y[S]\,Z[\cdot\cdot]$

(push-1) $\qquad$ $X[A\,\cdot\cdot] \xrightarrow{w} X[BC\,\cdot\cdot]$

(push-2) $\qquad$ $X[A\,\cdot\cdot] \xrightarrow{w} X[BC\,\cdot\cdot]$

Figure 4: Deductive systems for computing stringsums and allsums of CFG ▷ CFG.

form $(d_1, d_2)$, where

$$d_1 = X[A] \xRightarrow{p} X[BC] \xRightarrow{*} \boldsymbol{s}_{(i:j)} Y[C]\boldsymbol{\Upsilon}_2$$
$$\xRightarrow{*} \boldsymbol{s}_{(i:j)}\boldsymbol{s}_{(j:k)} Z[\varepsilon]\boldsymbol{\Upsilon}_1\boldsymbol{\Upsilon}_2,$$
$$d_2 = \boldsymbol{\Upsilon}_1\boldsymbol{\Upsilon}_2 \xRightarrow{*} \boldsymbol{s}_{(l:m)}\boldsymbol{\Upsilon}_2 \xRightarrow{*} \boldsymbol{s}_{(l:m)}\boldsymbol{s}_{(m:o)}.$$

By the inductive hypothesis, the weight $w_1$ of item is the weight of all pop computations ($X[B] \xRightarrow{*} \boldsymbol{s}_{(i:j)} Y[\varepsilon]\boldsymbol{\Upsilon}_2$, $\boldsymbol{\Upsilon}_2 \xRightarrow{*} \boldsymbol{s}_{(m:o)}$). Sim-

ilarly, the weight $w_2$ of item is the total weight of all pop computations ($Y[C] \xRightarrow{*} \boldsymbol{s}_{(j:k)} Z[\varepsilon]\boldsymbol{\Upsilon}_1$, $\boldsymbol{\Upsilon}_1 \xRightarrow{*} \boldsymbol{s}_{(l:m)}$). By distributivity, the total weight of all such partial derivations is $\mathbf{w}(p) \otimes w_1 \otimes w_2$. This weight is added to the weight of the item in inference rule (push-1).

Any pop computation starting with $X[A]$ and a

push production $p$ has the form

$$X[A] \overset{p}{\Rightarrow} X[BC] \overset{*}{\Rightarrow} \boldsymbol{s}_{(i:j)} Y[C] \Upsilon_2$$
$$\overset{*}{\Rightarrow} \boldsymbol{s}_{(i:j)} \boldsymbol{s}_{(j:k)} \Upsilon_2$$
$$\overset{*}{\Rightarrow} \boldsymbol{s}_{(i:j)} \boldsymbol{s}_{(j:k)} \boldsymbol{s}_{(k:l)}.$$

By the inductive hypothesis, the weight $w_1$ of item

is the weight of all pop computations $(X[B] \overset{*}{\Rightarrow} \boldsymbol{s}_{(i:j)} Y[\varepsilon] \Upsilon_2, \Upsilon_2 \overset{*}{\Rightarrow} \boldsymbol{s}_{(k:l)})$, while the weight $w_2$ of item

is the weight of all pop computations $Y[C] \overset{*}{\Rightarrow} \boldsymbol{s}_{(j:k)}$. By distributivity, the total weight of all such partial derivations is $\mathbf{w}(p) \otimes w_1 \otimes w_2$. This weight is added to the weight of the item

in inference rule (push-2).

$\square$

### 4.4 Complexity Analysis

Let $\mathcal{X}_1$ and $\mathcal{X}_2$ be the sets of nonterminals or stack symbols of the controllers and of the controllees, respectively. An algorithm based on one of our deductive systems would would need to store a

weight for each item

and

, for all $i, j, k, l \in [0:n]$, $A \in \mathcal{X}_1$, and $X, Y \in \mathcal{X}_2$, giving a space complexity of $\mathcal{O}(n^4 |\mathcal{X}_1||\mathcal{X}_2|^2)$ and $\mathcal{O}(n^2 |\mathcal{X}_1||\mathcal{X}_2|)$, respectively. Overall we get a space complexity of $\mathcal{O}(n^4 |\mathcal{X}_1|^2|\mathcal{X}_2|)$.

Computing the weight of each new item requires in the worst case inference rule (push-1), iterating over indices $i, j, k, l, m, n \in [0:n]$, and symbols $X, Y, Z \in \mathcal{X}_2$ and $A, B, C \in \mathcal{X}_1$. This gives a runtime of $\mathcal{O}(n^6 |\mathcal{X}_1|^3|\mathcal{X}_2|^3)$.

### 4.5 Comparison with Existing Algorithms

Vijay-Shanker and Weir (1989) designed a recognition (which is essentially computing stringsums in the boolean semiring) algorithm for a type of LIG that is d-strongly equivalent to a CFG in Chomsky normal form controlled by a simple PDA. Our stringsum algorithm is more space-efficient than theirs by a factor of $|\Gamma|$ and more time-efficient by a factor of $n|\mathcal{N}|$, where $n$ is the string length, $|\Gamma|$ is the size of the stack alphabet, and $|\mathcal{N}|$ is the size of the alphabet of nonterminals. Their algorithm could be improved using a trick similar to

the one used by Butoi et al. (2022), resulting in an algorithm with the same runtime as ours. But, in order to do so, it is necessary to store additional items, which increases the space complexity by another factor of $|\Gamma|$. Additionally, on the type of LIG used by Vijay-Shanker and Weir, we get a further runtime improvementof a factor of $\mathcal{O}(|\Gamma|)$.

For EPDA, our stringsum algorithm is more space-efficient than the algorithm of Alonso et al. (2001) by a factor of $|\Gamma|^2$ and has an improved runtime by a factor of $|\Gamma|^3$, where $|\Gamma|$ is the size of the stack alphabet. Their algorithm is designed for an EPDA without finite-state control, which is equivalent to a PDA $\triangleright$ PDA where both the controller and the controllee are single-state. Therefore, we exclude the states when comparing the two algorithms.

## 5 Computing Allsums

We reuse the notion of pop computation that we defined in §4.1 in order to derive a space-efficient algorithm for computing allsums of CFG $\triangleright$ CFG. Allsums of PDA $\triangleright$ CFG, CFG $\triangleright$ PDA, and PDA $\triangleright$ PDA can be computed similarly (see App. D). We define a new type of pop computation that will help us compute the weight of *all* derivations instead of derivations of a particular string.

**Definition 11.** *Let* $\mathbf{G}_1$ *be a controller WCFG and* $\mathbf{G}_2$ *a WLD-CFG, with nonterminal alphabets* $\mathcal{N}_1$ *and* $\mathcal{N}_2$, *respectively. Let be a symbol not occurring in the rules of* $\mathbf{G}_1 \triangleright \mathbf{G}_2$. *A **pop computation** of type*

, *where* $A \in \mathcal{N}_1$ *and* $X, Y \in \mathcal{N}_2$, *is a pair of partial derivations* $(d_1, d_2)$, *where*

$$d_1 \in (X[A] \overset{*}{\Rightarrow} \boldsymbol{s}_1 Y[\varepsilon] \Upsilon_2)$$
$$d_2 \in (\Upsilon_2 \overset{*}{\Rightarrow} \boldsymbol{s}_2)$$

*where* $\boldsymbol{s}_1, \boldsymbol{s}_2 \in \Sigma^*$ *and* $\Upsilon_2 \in \mathcal{N}_1[\mathcal{N}_2^*]^*$. *The weight of this pop computation is* $\mathbf{w}(d_1, d_2) = \mathbf{w}(d_1) \otimes \mathbf{w}(d_2)$.

*A **pop computation** of type* 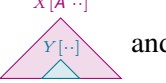 *is a partial derivation* $X[A] \overset{*}{\Rightarrow} \boldsymbol{s}$, *where* $\boldsymbol{s} \in \Sigma^*$.

Fig. 4b shows the deduction system for computing allsums. Just as for stringsums, we distinguish two types of items, those of type

and

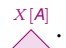.

**Theorem 2.** *Let $\mathbf{G}_1$ be a controller WCFG with nonterminal alphabet $\mathcal{N}_1$ and start symbol $S$, and $\mathbf{G}_2$ a WLD-CFG with nonterminal alphabet $\mathcal{N}_2$. Assume that both have weights in an $\omega$-continuous semiring (App. A.1). Then the allsum of $\mathbf{G}_1 \triangleright \mathbf{G}_2$ is the value of the item $\overset{S[S]}{\triangle}$ in the deductive system of Fig. 4b.*

Unlike for stringsums, this deduction system has cycles. The value of the goal item $\overset{S[S]}{\triangle}$ can be found using fixed-point iteration (Goodman, 1999) or the semiring generalization of Newton's method (Esparza et al., 2007).

Since there are various algorithms for computing item values, we don't analyze time complexity, but only space complexity. Let $\mathcal{N}_1$ and $\mathcal{N}_2$ be the sets of nonterminals or stack symbols of the controller and the controllee, respectively. As the algorithm needs to store an item of the form $\overset{X[A\,\cdot\cdot]}{\underset{Y[\cdot\cdot]}{\triangle}}$ for each $X, Y \in \mathcal{N}_2$ and $A \in \mathcal{N}_1$, and an item of the form $\overset{X[A]}{\triangle}$ for each $X \in \mathcal{N}_2$ and $A \in \mathcal{N}_1$, it has a space complexity of $\mathcal{O}(|\mathcal{N}_1||\mathcal{N}_2|^2)$. If we were to compute allsums in LIG using an algorithm based on Vijay-Shanker and Weir's (1989) algorithm, we would have stored $\mathcal{O}(|\mathcal{N}_1|^2|\mathcal{N}_2|^2)$ items, therefore we have a space improvement of a factor of $|\mathcal{N}_1|$. Similarly, we get a space improvement for EPDA of a factor of $|\mathcal{N}_1|^2$ over the allsum algorithm based on Alonso et al.'s (2001) algorithm.

## 6 Conclusion

Our work has contributed several new results and algorithms for $\mathcal{L}_2$ formalisms. We introduced semiring-weighted versions of controllable CFGs and PDAs, which give rise naturally to four semiring-weighted two-level formalisms when they are controlled by semiring-weighted CFGs and PDAs. We also introduced a WPDA normal form that is completely analogous to the Chomsky normal form for CFGs and showed that one can derive normal forms for the two-level formalisms only from the normal forms of the controller and of the controllee. These normal forms are also normal forms for TAG, LIG, PAA, and EPDA, respectively, and the conversions only require conversions of the controller and the controllee. Finally, we designed new stringsum and allsum algorithms for all of these formalisms, some of which are faster or more space-efficient than several existing algorithms.

## Limitations

Our stringsum algorithms can be used for LIG, EPDA, spinal TAG, and spinal PAA by first converting them to a two-level formalism, converting the resulting controller and controllee grammar/automaton into the normal form, and then merging their rules into a single set of rules. Similarly, for a general two-level formalism, the rules of the controller and the controllee would have to be extracted from the merged rules, converted into the normal form, and merged back before using the stringsum or allsum algorithm. Although simpler, requiring only CFG and PDA conversions, these transformations add some extra complexity. We leave direct normal form conversions for future work.

## Ethics Statement

The authors foresee no ethical concerns with the research presented in this paper.

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

# A   Additional Preliminaries

## A.1   Semirings

**Definition 12.** *A **monoid** is a tuple $(W, \odot, \boldsymbol{I})$, where $W$ is a set, $\odot$ is an associative binary operation, and $\boldsymbol{I} \in W$, called the **identity** element, satisfies $\boldsymbol{I} \odot a = a \odot \boldsymbol{I} = a$ for all $a \in W$. If $a \odot b = b \odot a$ for all $a, b$, we say that the monoid is **commutative**.*

**Definition 13.** *A **semiring** is a tuple $\mathcal{W} = (W, \oplus, \otimes, \boldsymbol{0}, \boldsymbol{1})$ such that $(W, \oplus, \boldsymbol{0})$ is a commutative monoid and $(W, \otimes, \boldsymbol{1})$ is a monoid. Additionally, $\otimes$ distributes over $\oplus$, that is, $a \otimes (b \oplus c) = a \otimes b \oplus a \otimes c$ and $(a \oplus b) \otimes c = a \otimes c \oplus b \otimes c$, and $\boldsymbol{0}$ is absorbing with respect to $\otimes$, that is, $\boldsymbol{0} \otimes a = a \otimes \boldsymbol{0} = \boldsymbol{0}$. If $\otimes$ is commutative, then we say that $\mathcal{W}$ is **commutative**.*

The following definitions are from Esparza et al. (2007):

**Definition 14.** *If $\mathcal{W} = (W, \oplus, \otimes, \boldsymbol{0}, \boldsymbol{1})$ is a semiring, the **natural order** on $\mathcal{W}$ is the relation $\leq$, defined such that $a \leq b \Leftrightarrow \exists d \in W : a \oplus d = b$. If $\leq$ is a partial order (i.e., reflexive, antisymmetric, and transitive), then we say that $\mathcal{W}$ is **naturally ordered**.*

**Definition 15.** *A naturally ordered semiring $\mathcal{W} = (W, \oplus, \otimes, \boldsymbol{0}, \boldsymbol{1})$ is $\omega$-continuous if it is additionally equipped with an infinite summation operator $\bigoplus$ such that:*

- *For any countable sequence $(a_1, a_2, \ldots)$, the sequence $\left( \bigoplus_{i=1}^{k} a_i \right)_{k \geq 0}$ has a least upper bound in $W$ and is equal to $\bigoplus_i a_i$.*

- *Multiplication distributes over infinite sums:*

$$\bigoplus_i (c \otimes a_i) = c \otimes \left( \bigoplus_i a_i \right)$$
$$\bigoplus_i (a_i \otimes c) = \left( \bigoplus_i a_i \right) \otimes c.$$

- *For any partition $(I_1, I_2, \ldots)$ of $\mathbb{N}$,*

$$\bigoplus_j \bigoplus_{i \in I_j} a_i = \bigoplus_i a_i.$$

## A.2   Weighted Pushdown Automata

We use a definition of WPDA (Butoi et al., 2022) that is more general than usual and roughly a weighted version of the extended PDAs of Aho and Ullman (1972, p. 173) and the PDAs of Lewis and Papadimitriou (1997, p. 131).

**Definition 16.** *A **weighted pushdown automaton** (WPDA) over a semiring $\mathcal{W} = W$ is a tuple $\mathbf{P} = (Q, \Sigma, \Gamma, \delta, \mathrm{w}, (q_\iota, \boldsymbol{\gamma}_\iota), (q_f, \boldsymbol{\gamma}_f))$, where*

- *$Q$, $\Sigma$, and $\Gamma$ are finite sets of states, input symbols, and stack symbols, respectively,*

- *$\delta \subseteq Q \times \Gamma^* \times (\Sigma \cup \{\varepsilon\}) \times Q \times \Gamma^*$ is a finite set of transitions,*

- *$\mathrm{w} \colon \delta \to W$ is a transition-weighting function, and*

- *$(q_\iota, \boldsymbol{\gamma}_\iota)$ and $(q_f, \boldsymbol{\gamma}_f)$, where $q_\iota, q_f \in Q, \boldsymbol{\gamma}_\iota, \boldsymbol{\gamma}_f \in \Gamma^*$, are called the initial and final configurations.*

*If $(p, \boldsymbol{\gamma}, a, q, \boldsymbol{\gamma}') \in \delta$ is a transition, we write it as $p, \boldsymbol{\gamma} \xrightarrow{a} q, \boldsymbol{\gamma}'$. If $\mathrm{w}(p, \boldsymbol{\gamma} \xrightarrow{a} q, \boldsymbol{\gamma}') = w$, we use the notation $p, \boldsymbol{\gamma} \xrightarrow{a/w} q, \boldsymbol{\gamma}'$.*

We treat strings in $\Gamma^*$ as stacks, with the first symbol being the top of the stack and the last symbol being the bottom.

A WPDA moves from one configuration to another by following transitions of the form $(p, \boldsymbol{\gamma} \xrightarrow{a} q, \boldsymbol{\gamma}')$, which represents a move from state $p$ to state $q$, while scanning the symbol $a$, popping the sequence $\boldsymbol{\gamma}$ from the top of the stack and pushing the sequence $\boldsymbol{\gamma}'$. In the following definitions, let $\mathbf{P} = (Q, \Sigma, \Gamma, \delta, \mathrm{w}, (q_\iota, \boldsymbol{\gamma}_\iota), (q_f, \boldsymbol{\gamma}_f))$ be a WPDA.

**Definition 17.** *A **configuration** of $\mathbf{P}$ is a pair $(q, \boldsymbol{\gamma})$, where $q \in Q$ is the current state and $\boldsymbol{\gamma} \in \Gamma^*$ is the current contents of the stack. We write $\mathcal{C}(\mathbf{P})$ for the set of all configurations of $\mathbf{P}$.*

**Definition 18.** *If $(p, \boldsymbol{\gamma}\boldsymbol{\alpha})$ and $(q, \boldsymbol{\gamma}'\boldsymbol{\alpha})$ are configurations of $\mathbf{P}$ and $\tau = (p, \boldsymbol{\gamma} \xrightarrow{a} w, q\boldsymbol{\gamma}')$ is a transition of $\mathbf{P}$, we write $(p, \boldsymbol{\gamma}\boldsymbol{\alpha}) \xRightarrow{\tau} (q, \boldsymbol{\gamma}'\boldsymbol{\alpha})$ to denote that configuration $(q, \boldsymbol{\gamma}'\boldsymbol{\alpha})$ can be reached from $(p, \boldsymbol{\gamma}\boldsymbol{\alpha})$ in one step using transition $\tau$.*

**Definition 19.** *We say that a transition $\tau$ is $k$-pop, $l$-push if $|\boldsymbol{\gamma}| = k$ and $|\boldsymbol{\gamma}'| = l$. We say that $\tau$ **scans** $a$, and if $a \neq \varepsilon$, we call $\tau$ **scanning**; otherwise, we call it **non-scanning**.*

The following type of WPDA is a weighted version of the PDA used by Lang (1974) for his recognition algorithm.

**Definition 20.** *A WPDA is called **simple** if each of its transitions is $k$-pop, $l$-push for some $k \leq 1$ and $l \leq 1$.*

The following type of WPDA is a weighted version of the PDA used by (Hopcroft et al., 2006) and others.

**Definition 21.** *A WPDA is called **top-down** if all of its transitions are $1$-pop, and the initial and final configurations are $(q_\iota, S)$ and $(q_f, \varepsilon)$, respectively.*

**Definition 22.** *A **run** of $\mathbf{P}$ from $(q_0, \boldsymbol{\gamma}_0)$ to $(q_n, \boldsymbol{\gamma}_n)$ is a sequence of steps $\boldsymbol{\pi} = (q_0, \boldsymbol{\gamma}_0) \xRightarrow{\tau_1} \cdots \xRightarrow{\tau_n} (q_n, \boldsymbol{\gamma}_n)$. We write $(q_0, \boldsymbol{\gamma}_0) \xRightarrow{*} (q_n, \boldsymbol{\gamma}_n)$ to assert that a run from $(p, \boldsymbol{\gamma})$ to $(q, \boldsymbol{\gamma}')$ exists, or to denote the set of all such runs. If $(q_0, \boldsymbol{\gamma}_0) = (q_\iota, \boldsymbol{\gamma}_\iota)$ and $(q_n, \boldsymbol{\gamma}_n) = (q_f, \boldsymbol{\gamma}_f)$, we call $\boldsymbol{\pi}$ **accepting**. If $\tau_i$ scans $a_i$, for $i \in [1{:}n]$, then we say that $\boldsymbol{\pi}$ scans $\boldsymbol{s} = a_1 \cdots a_n$.*

*The **weight** of $\boldsymbol{\pi}$ is the product of its transition weights,*

$$\mathrm{w}(\boldsymbol{\pi}) \overset{\text{def}}{=} \bigotimes_{i=1}^{n} \mathrm{w}(\tau_i).$$

*We denote by $\Pi(\mathbf{P}, \boldsymbol{s})$ the set all accepting runs of $\mathbf{P}$ scanning $\boldsymbol{s}$, and by $\Pi(\mathbf{P})$ the set of all accepting runs of $\mathbf{P}$.*

**Definition 23.** *The **stringsum** $\mathrm{w}(\mathbf{P}, \boldsymbol{s})$ of a string $\boldsymbol{s}$ under $\mathbf{P}$ is the total weight of all accepting runs scanning $\boldsymbol{s}$,*

$$\mathrm{w}(\mathbf{P}, \boldsymbol{s}) \overset{\text{def}}{=} \bigoplus_{\boldsymbol{\pi} \in \Pi(\mathbf{P}, \boldsymbol{s})} \mathrm{w}(\boldsymbol{\pi}).$$

**Definition 24.** *The **allsum** $\mathrm{w}(\mathbf{P})$ of $\mathbf{P}$ is the total weight of all its accepting runs,*

$$\mathrm{w}(\mathbf{P}) \overset{\text{def}}{=} \bigoplus_{\boldsymbol{\pi} \in \Pi(\mathbf{P})} \mathrm{w}(\boldsymbol{\pi}).$$

## B  Two-Level Formal Systems

### B.1  Definitions

Having defined both WPDA (Def. 16) and WLD-CFG (Def. 7), we can give a definition for weighted PDA $\triangleright$ CFG.

**Definition 25.** *Let $\mathbf{P}_1$ be a controller WPDA with states $Q$, stack alphabet $\Gamma_1$ and initial configuration $(q_\iota, \boldsymbol{\gamma}_\iota)$, and let $\mathbf{G}_2$ be a controllee WLD-CFG with nonterminals $\mathcal{N}_2$. Then $\mathbf{P}_1 \triangleright \mathbf{G}_2$ is a rewriting system with rules as follows.*

(a) If $(q, \mathbf{A} \xrightarrow{\varepsilon/w} r, \boldsymbol{\gamma})$ is a transition of $\mathbf{P}_1$, then $\mathbf{P}_1 \triangleright \mathbf{G}_2$ has a rule for each $X \in \mathcal{N}_2$:

$$X[q, \mathbf{A}\cdots] \xrightarrow{w} X[r, \boldsymbol{\gamma}\cdots].$$

(b) If $(r, \mathbf{A} \xrightarrow{\ell/w_1} s, \varepsilon)$ is a transition of $\mathbf{P}_1$, and $(\ell: X \xrightarrow{w_2} \boldsymbol{\alpha}_1 \breve{Y} \boldsymbol{\alpha}_2)$ is a production of $\mathbf{G}_2$, then $\mathbf{G}_1 \triangleright \mathbf{G}_2$ has a rule

$$X[r, \mathbf{A}\cdots] \xrightarrow{w_1 \otimes w_2} \boldsymbol{\alpha}_1[q_\iota, \boldsymbol{\gamma}_\iota] \, Y[s, \cdots] \, \boldsymbol{\alpha}_2[q_\iota, \boldsymbol{\gamma}_\iota].$$

Next, we define WLD-PDAs by analogy with WLD-CFGs. This definition is a weighted version of our previous definition (Butoi et al., 2023).

**Definition 26.** *A **weighted labeled distinguished pushdown automaton** (WLD-PDA) over a semiring $\mathcal{W} = (W, \oplus, \otimes, \mathbf{0}, \mathbf{1})$ is a tuple $\mathbf{P} = (Q, \Sigma, \Gamma, L, \delta, \mathrm{w}, (q_\iota, S), (q_f, \varepsilon))$, where*

- *$Q$, $\Sigma$, $\Gamma$ and $L$ are finite sets of states, input symbols, stack symbols and labels, respectively,*

- *$\delta \subseteq L \times \mathbb{N} \times Q \times \Gamma \times (\Sigma \cup \{\varepsilon\}) \times Q \times \Gamma^*$ is a finite set of transitions,*

- *$\mathrm{w} \colon \delta \to W$ is the transition-weighting function, and*

- *$(q_\iota, S)$ and $(q_f, \varepsilon)$, where $q_\iota, q_f \in Q, S \in \Gamma$, are the initial and final configurations.*

*If $(\ell, i, q, \mathbf{A}, a, r, \mathbf{B}_1 \cdots \mathbf{B}_n)$ is a transition in $\delta$, we must have either $i = 0$, which we write as $q, \mathbf{A} \xrightarrow{a} r, \mathbf{B}_1 \cdots \mathbf{B}_n$, or $1 \le i \le n$, which we write as $q, \mathbf{A} \xrightarrow{a} r, \mathbf{B}_1 \cdots \mathbf{B}_{i-1} \breve{\mathbf{B}}_i \mathbf{B}_{i+1} \cdots \mathbf{B}_n$.*

A WLD-PDA behaves similarly to a WPDA, but its runs are controlled by either a controller CFG or PDA. As in §3, the rules of the controller and of the controllee can be merged into a single system.

**Definition 27.** *Let $\mathbf{G}_1$ be a controller CFG with nonterminals $\mathcal{N}$ and start symbol $\mathsf{S}$, and let $\mathbf{P}_2$ be a WLD-PDA with stack alphabet $\Gamma$ and states $Q$. Then $\mathbf{G}_1 \triangleright \mathbf{P}_2$ is a rewriting system with rules as follows.*

(a) *If $(\mathbf{A} \xrightarrow{w} \boldsymbol{\beta})$ is a production of $\mathbf{G}_1$, where $\boldsymbol{\beta} \in \mathcal{N}^*$, then $\mathbf{G} \triangleright \mathbf{P}_2$ has a transition for each $X \in \Gamma$ and $q \in Q$:*

$$q, X[\mathbf{A}\cdots] \xrightarrow{\varepsilon/w} q, X[\boldsymbol{\beta}\cdots].$$

(b) *If $(\mathbf{A} \xrightarrow{w_1} \ell)$ is a production of $\mathbf{G}_1$, and $\ell: q, X \xrightarrow{a/w_2} r, \boldsymbol{\alpha}_1 \breve{Y} \boldsymbol{\alpha}_2$ is a transition of $\mathbf{P}_2$, then $\mathbf{G} \triangleright \mathbf{P}_2$ has a transition*

$$q, X[\mathbf{A}\cdots] \xrightarrow{a/w_1 \otimes w_2} r, \boldsymbol{\alpha}_1[\mathsf{S}] \, Y[\cdots] \, \boldsymbol{\alpha}_2[\mathsf{S}].$$

**Definition 28.** *Let $\mathbf{P}_1$ be a controller PDA with stack alphabet $\Gamma_1$ and start configuration $(q_\iota, \boldsymbol{\gamma}_\iota)$, and let $\mathbf{P}_2$ be a WLD-PDA with stack alphabet $\Gamma_2$ and states $Q_2$. Then $\mathbf{P}_1 \triangleright \mathbf{P}_2$ is a rewriting system with rules as follows.*

(a) *If $(q, \mathbf{A} \xrightarrow{\varepsilon/w} r, \boldsymbol{\gamma})$ is a transition of $\mathbf{P}_1$, then $\mathbf{P}_1 \triangleright \mathbf{P}_2$ has a transition for each $X \in \mathcal{N}_2$ and $p \in Q_2$:*

$$p, X[q, \mathbf{A}\cdots] \xrightarrow{\varepsilon/w} p, X[r, \boldsymbol{\gamma}\cdots].$$

(b) *If $(r, \mathbf{A} \xrightarrow{\ell/w_1} s, \varepsilon)$ is a transition of $\mathbf{P}_1$, and $\ell: p, X \xrightarrow{a/w_2} q, \boldsymbol{\alpha}_1 \breve{Y} \boldsymbol{\alpha}_2$ is a transition of $\mathbf{P}_2$, then $\mathbf{G} \triangleright \mathbf{P}_2$ has a transition*

$$p, X[r, \mathbf{A}\cdots] \xrightarrow{a/w_1 \otimes w_2} q, \boldsymbol{\alpha}_1[q_\iota, \boldsymbol{\gamma}_\iota] \, Y[s, \cdots] \boldsymbol{\alpha}_2[q_\iota, \boldsymbol{\gamma}_\iota].$$

The terms scanning/non-scanning transitions and (accepting) runs are defined analogously to those for WPDAs.

## B.2 Normal Forms

### B.2.1 WCFG normal form

**Proposition 1.** *For any WCFG $\mathbf{G}_1$ with weights in an $\omega$-continuous semiring, there is a WCFG in Chomsky normal form that defines the same weighted language as $\mathbf{G}_1$.*

*For any WLD-CFG $\mathbf{G}_2$ with weights in an $\omega$-continuous semiring, there is a WLD-CFG in Chomsky normal form that is equivalent to $\mathbf{G}_2$.*

*Proof.* The proof for WCFGs is a generalization of the standard conversion for unweighted grammars (Hopcroft et al., 2006), and is related to the Earley-style algorithm of Stolcke (1995).

1. If $S$ is the old start symbol, add a new nonterminal $S'$, a production $S' \xrightarrow{1} S$, and make $S'$ the new start symbol.

2. For every terminal $a$, create a production $T_a \xrightarrow{1} a$, where $T_a$ is a fresh nonterminal, and replace every right-hand side occurrence of $a$ with $T_a$.

3. Replace every production $X \xrightarrow{w} Y_1 \cdots Y_k$, where $k > 2$, with the productions

$$X \xrightarrow{w} Y_1 Z_1$$
$$Z_1 \xrightarrow{1} Y_2 Z_2$$
$$\vdots$$
$$Z_{k-2} \xrightarrow{1} Y_{k-1} Y_k$$

   where $Z_1, \ldots, Z_{k-2}$ are fresh nonterminals.

4. For each nonterminal $X$, compute the weight of all partial derivations $X \overset{*}{\Rightarrow} \varepsilon$ using the deductive system

$$\frac{}{A \overset{*}{\Rightarrow} \varepsilon} \quad A \xrightarrow{w} \varepsilon$$

$$\frac{B \overset{*}{\Rightarrow} \varepsilon}{A \overset{*}{\Rightarrow} \varepsilon} \quad A \xrightarrow{w} B$$

$$\frac{B \overset{*}{\Rightarrow} \varepsilon \quad C \overset{*}{\Rightarrow} \varepsilon}{A \overset{*}{\Rightarrow} \varepsilon} \quad A \xrightarrow{w} BC$$

   Then for every production $X \xrightarrow{w} YZ$ with $\mathbf{w}(Y \overset{*}{\Rightarrow} \varepsilon) = w_Y$ and $\mathbf{w}(Z \overset{*}{\Rightarrow} \varepsilon) = w_Z$, add productions $X \xrightarrow{w_1 \otimes w_Y} Z$ and $X \xrightarrow{w_1 \otimes w_Z} Y$. Finally, for all $X$, remove all productions $X \to \varepsilon$.

5. For all nonterminals $X$ and $Y$, compute the weight of all partial derivations $X \overset{*}{\Rightarrow} Y$ using the deductive system

$$\frac{}{A \overset{*}{\Rightarrow} A}$$

$$\frac{A \overset{*}{\Rightarrow} B}{A \overset{*}{\Rightarrow} C} \quad B \xrightarrow{w} C$$

   Then for every production $X \xrightarrow{w_2} YZ$ and nonterminal $W \neq X$, with $\mathbf{w}(W \overset{*}{\Rightarrow} X) = w_1$, add production $W \xrightarrow{w_1 \otimes w_2} YZ$. Finally, for all $X, Y$, remove all productions $X \to Y$.

Now we give a similar construction for WLD-CFGs. Let $\mathbf{G}$ be a WLD-CFG and $\mathbf{F}$ a controller WCFG or WPDA, both with weights from an $\omega$-continuous semiring. Then there exists a WLD-CFG in normal form $\mathbf{G}'$ and a controller $\mathbf{F}'$ such that $\mathbf{F}' \triangleright \mathbf{G}'$ defines the same weighted language as $\mathbf{F} \triangleright \mathbf{G}$. We assume for simplicity that $\mathbf{F}$ is a controller WCFG; a similar construction can be given when it is a WPDA.

Converting a CFG ▷ CFG to normal form requires an allsum algorithm that works on any binarized CFG ▷ CFG, not necessarily in normal form. We must use the rules shown in Fig. 5 in addition to those from Fig. 4b when computing allsums.

(nullary)

$$\frac{}{X[A]} \qquad\qquad X[A] \xrightarrow{w} \varepsilon$$

(controllee unary)

$$\frac{}{X[A\cdots]} \qquad X[A\cdots] \xrightarrow{w} Y[\cdot\cdot]$$

(controller unary)

$$\frac{X[B\cdots]}{X[A\cdots]} \qquad X[A\cdots] \xrightarrow{w} X[B\cdots]$$

Figure 5: Additional deduction rules for computing allsums in binarized CFG ▷ CFG.

**Root–foot transform** The nullary and unary removal steps below are analogous to nullary and unary removal in a WCFG. However, sometimes, when we remove nullary/unary rules, we need information from both the controller and controllee to compute their lost weight. To facilitate this, we define the following transformation, which copies information from the controllee to the controller.

Let $\mathcal{N}_1$ and $\mathcal{N}_2$ be the controller and controllee nonterminal alphabet, respectively.

1. For every controllee rule $\ell : X \to \boldsymbol{\alpha}$ where $\boldsymbol{\alpha}$ has no distinguished symbol, relabel the rule as $\ell_{\perp}^X$.

2. For every controllee rule $\ell : X \to \boldsymbol{\alpha} \breve{Y} \boldsymbol{\beta}$, relabel the rule as $\ell_Y^X$.

3. Perform the construction of Bar-Hillel et al. (1961) on the controller, using controllee nonterminals instead of states. That is, relabel every controller nonterminal $A$ as $A_{\perp}^X$ or $A_Y^X$ for all $X, Y \in \mathcal{N}_2$, making as many copies of rules as needed, subject to the constraints that (i) if the lhs is $A_Y^X$ (where $Y$ could be $\perp$) then the first rhs symbol must be $A_{Y'}^X$, and the last rhs symbol must be $A_Y^{X'}$; (ii) if $A_Y^X$ is an rhs nonterminal then its successor (if any) must be $B_Z^Y$.

4. Add a new controller start symbol $S'$ with rules $S' \to S_{\perp}^X$ for all $X \in \mathcal{N}_2$.

**Controllee binarization** In the controllee, replace every production $\ell : X \xrightarrow{w} Y_1 \cdots \breve{Y}_d \cdots Y_k$, where $k > 2$, with the productions

$$\ell_1 : X \xrightarrow{w} Y_1 \breve{Z}_1$$
$$\ell_2 : Z_1 \xrightarrow{\mathbf{1}} Y_2 \breve{Z}_2$$
$$\vdots$$
$$\ell_{d-1} : Z_{d-2} \xrightarrow{\mathbf{1}} Y_{d-1} \breve{Z}_d$$
$$\ell_d : Z_d \xrightarrow{\mathbf{1}} \widetilde{Z_{d+1}} Y_k$$
$$\ell_{d+1} : Z_{d+1} \xrightarrow{\mathbf{1}} \widetilde{Z_{d+2}} Y_{k-1}$$
$$\vdots$$
$$\ell_{k-1} : Z_{k-1} \xrightarrow{\mathbf{1}} \breve{Y}_d Y_{d+1}$$

where $Z_1, \ldots, Z_{k-2}$ are fresh controllee nonterminals.

Accordingly, in the controller, replace every rhs occurrence of $\ell$ with $\ell_1 \cdots \ell_{k-1}$.

**Controllee nullary removal** Like the standard nullary removal algorithm (Hopcroft et al., 2006), nullary removal in an WLD-CFG has three steps: partitioning, precomputation, and removal. However, removing nullary rules from the controllee also sometimes requires modifications to the controller.

1. Partition: Replace every controllee nonterminal $X$ with two nonterminals $X_\varepsilon$ and $X_{\not\varepsilon}$. Every rule with $k$ rhs nonterminals becomes $2^k$ rules with all possible combination of rhs nonterminals, and the lhs annotated $\varepsilon$ iff the rhs is empty or all the rhs symbols are annotated $\varepsilon$. The start symbol is $S_{\not\varepsilon}$.

2. Precompute: Compute the allsum $\mathbf{w}\left( \begin{smallmatrix} X_\varepsilon[A] \\ \triangle \end{smallmatrix} \right)$ for all $A \in \mathcal{N}_1, X_\varepsilon \in \mathcal{N}_2$.

3. Remove (non-distinguished): For every non-distinguished occurrence of $X_\varepsilon$ in the rhs of a controllee rule $\pi$, delete the occurrence and multiply the weight of $\pi$ by $\mathbf{w}\left( \begin{smallmatrix} X_\varepsilon[S] \\ \triangle \end{smallmatrix} \right)$.

4. Remove (distinguished): The distinguished occurrences of $X_\varepsilon$ cannot be removed in the same way, because we don't know what weight to multiply in. Instead, perform the root–foot transform. Then, for every occurrence of $A_\perp^{X_\varepsilon}$ on the rhs of a controller rule $\pi$, delete the occurrence and multiply the weight of $\pi$ by $\mathbf{w}\left( \begin{smallmatrix} X_\varepsilon[A] \\ \triangle \end{smallmatrix} \right)$.

5. Remove (start): Add controller rule $S' \xrightarrow{1} \ell$ and controllee rule $\ell : S^{\not\varepsilon} \xrightarrow{w} \varepsilon$, where $w = \mathbf{w}\left( \begin{smallmatrix} S_\varepsilon[S] \\ \triangle \end{smallmatrix} \right)$.

6. Remove every controller rule $S' \to \varepsilon$.

7. Remove every controller rule with lhs $A_\perp^{X_\varepsilon}$ and every controllee rule with lhs $X_\varepsilon$.

**Controllee unary removal**

1. Partition: Replace every controller nonterminal $A$ with two nonterminals $A_\mathsf{U}$ and $A_{\not\mathsf{U}}$. Every rule with $k$ rhs nonterminals becomes $2^k$ rules. A rule's lhs is annotated $\mathsf{U}$ iff all of its rhs symbols are either labels of controllee unary rules or annotated $\mathsf{U}$. The controller temporarily has two start symbols, $S_\mathsf{U}$ and $S_{\not\mathsf{U}}$.

2. Precompute: Compute the allsum $\mathbf{w}\left( \begin{smallmatrix} X[A_\mathsf{U} \,\cdot\cdot] \\ \triangle \\ Y[\cdot\cdot] \end{smallmatrix} \right)$ for all $A_\mathsf{U} \in \mathcal{N}_1, X, Y \in \mathcal{N}_2$.

3. Remove (controller): Perform the root–foot transform. Then, for every occurrence of $A_\mathsf{U}{}_Y^X$ on the rhs of rule $\pi$, delete the occurrence and multiply the weight of $\pi$ by $\mathbf{w}\left( \begin{smallmatrix} A_\mathsf{U}[X \,\cdot\cdot] \\ \triangle \\ Y[\cdot\cdot] \end{smallmatrix} \right)$. Remove all rules with lhs $X_Y^{A_\mathsf{U}}$.

4. Remove (controllee): Remove all unary rules, and for each rule $\ell : Y \xrightarrow{w} \boldsymbol{\alpha}$, and all $X \in \mathcal{N}_2$, make a copy $\ell : X \xrightarrow{w} \boldsymbol{\alpha}$.

$\square$

### B.2.2 WPDA normal form

We first define a WPDA normal form that is analogous to Chomsky normal form for WCFGs.

**Definition 29.** *A WPDA is in **normal form** if all of its transitions are of one of the following types: (1) $q_\iota, S \xrightarrow{\varepsilon/w} q_f, \varepsilon$, (2) scanning and 0-push, or (3) non-scanning and 2-push. Moreover, $S$ is not pushed by any transition.*

*A WLD-PDA is in normal form if all of its transitions are of type (1) or (2) above, (3a) $q, X \xrightarrow{\varepsilon/w} r, \breve{Y} Z$, or (3b) $q, X \xrightarrow{\varepsilon/w} r, Y \breve{Z}$.*

**Proposition 2.** *Let $\mathbf{P}$ be a WPDA with weights from an $\omega$-continuous semiring. Then there is a WPDA in normal form that defines the same weighted language as $\mathbf{P}$.*

*For any WLD-PDA $\mathbf{P}_2$ with weights in an $\omega$-continuous semiring, there is a WLD-PDA in Chomsky normal form that is equivalent to $\mathbf{P}_2$.*

*Proof.* In previous work (Butoi et al., 2022), we gave a conversion from arbitrary WPDAs to a normal form that is close to Def. 29, but allows transitions that are scanning but are 1-push or 2-push. We can modify the conversion using the following preparatory steps:

1. If $(q_\iota, \boldsymbol{\gamma}_\iota)$ is the old start configuration, add a new state $q_\iota'$, a new stack symbol $S'$, and a transition $q_\iota', S' \xrightarrow{\mathbf{1}} q_\iota, \boldsymbol{\gamma}_\iota$, and make $(q_\iota', S')$ the new start configuration.

2. For every terminal $a$, add a fresh stack symbol $T_a$. Replace every scanning transition $(q, X \xrightarrow{a/w} r, \gamma)$ with $(q, X \xrightarrow{\epsilon/w} r, T_a\gamma)$, and add transitions $(q, T_a \xrightarrow{a/\mathbf{1}} q, \varepsilon)$ for every state $q$.

3. Remove all nullary transitions (Butoi et al., 2022, §3.2).

4. Remove all unary transitions (Butoi et al., 2022, §3.3).

A WLD-PDA can always be converted into an equivalent WLD-CFG (Butoi et al., 2023), converted into the normal form, then converted back into into a WLD-PDA. $\qquad\square$

To convert a WLD-PDA to normal form, we use the construction above, modified by analogy with Prop. 1. In the root–foot transform, each nonterminal includes two or four states (just as allsum items do). The nullary removal procedure for WPDAs is more complex than for WCFGs, but the required modification is analogous: we remove non-distinguished stack symbols $X_\varepsilon$ as usual, but when removing a distinguished stack symbol $X_\varepsilon$, we must multiply its weight in the controller rather than the controllee.

### B.2.3 Two-level normal forms

Fig. 6 shows how the normal forms of PDA ▷ CFG, CFG ▷ PDA and PDA ▷ PDA are obtained from the normal forms of the controllees and the controllers. For the epsilon rules, as noted in §3.2, in principle there could be rules with other left-hand sides, but such rules would never be used.

## C Computing Stringsums

### C.1 Pop Computations

**Definition 30.** *Let $\mathbf{P}$ be a controller WPDA with states $Q$, and stack alphabet $\Gamma$, and let $\mathbf{G}$ be a WLD-CFG with nonterminals $\mathcal{N}$. Let $\boldsymbol{s} \in \Sigma^*$ be a string of length $|\boldsymbol{s}| = n$. A **pop computation** of $\mathbf{P} \triangleright \mathbf{G}$ of type*

*, where $0 \le i \le j \le k \le l \le n$, $p, q \in Q$, $X, Y \in \mathcal{N}$, and $A \in \Gamma$, is a pair of partial derivations $(d_1, d_2)$, where*

$$d_1 \in (X[p, A] \overset{*}{\Rightarrow} \boldsymbol{s}_{(i:j)} Y[q, \varepsilon] \boldsymbol{\Upsilon}_2)$$

$$d_2 \in (\boldsymbol{\Upsilon}_2 \overset{*}{\Rightarrow} \boldsymbol{s}_{(k:l)}).$$

| $\mathbf{P}_1$ | $\mathbf{G}_2$ | $\mathbf{P}_1 \triangleright \mathbf{G}_2$ | name |
|---|---|---|---|
| $q_\iota, S \xrightarrow{\ell/w_1} q_f, \varepsilon$ | $\ell: S \xrightarrow{w_2} \varepsilon$ | $S[q_\iota, S] \xrightarrow{w_1 \otimes w_2} \varepsilon$ | (epsilon) |
| $p, A \xrightarrow{\ell/w_1} q_f, \varepsilon$ | $\ell: X \xrightarrow{w_2} a$ | $X[p, A] \xrightarrow{w_1 \otimes w_2} a$ | (terminal) |
| $p, A \xrightarrow{\ell/w_1} q, \varepsilon$ | $\ell: X \xrightarrow{w_2} \breve{Y} Z$ | $X[p, A\cdot\cdot] \xrightarrow{w_1 \otimes w_2} Y[q, \cdot\cdot]Z[q_\iota, S]$ | (left pop) |
| $p, A \xrightarrow{\ell/w_1} q, \varepsilon$ | $\ell: X \xrightarrow{w_2} Y \breve{Z}$ | $X[p, A\cdot\cdot] \xrightarrow{w_1 \otimes w_2} Y[q_\iota, S]Z[q, \cdot\cdot]$ | (right pop) |
| $p, A \xrightarrow{\varepsilon/w_1} q, BC$ | | $X[q, A\cdot\cdot] \xrightarrow{w_1} X[q, BC\cdot\cdot]$ | (push) |

(a) PDA $\triangleright$ CFG.

| $\mathbf{G}_1$ | $\mathbf{P}_2$ | $\mathbf{G}_1 \triangleright \mathbf{P}_2$ | name |
|---|---|---|---|
| $S \xrightarrow{w_1} \ell$ | $\ell: q_\iota, S \xrightarrow{\varepsilon/w_2} q_f, \varepsilon$ | $q_\iota, S[S] \xrightarrow{\varepsilon/w_1 \otimes w_2} q_f, \varepsilon$ | (epsilon) |
| $A \xrightarrow{w_1} \ell$ | $\ell: q, X \xrightarrow{a/w_2} r, \varepsilon$ | $q, X[A] \xrightarrow{a/w_1 \otimes w_2} r, \varepsilon$ | (terminal) |
| $A \xrightarrow{w_1} \ell$ | $\ell: q, X \xrightarrow{\varepsilon/w_2} r, \breve{Y} Z$ | $q, X[A\cdot\cdot] \xrightarrow{\varepsilon/w_1 \otimes w_2} r, Y[\cdot\cdot]Z[S]$ | (left pop) |
| $A \xrightarrow{w_1} \ell$ | $\ell: q, X \xrightarrow{\varepsilon/w_2} r, Y \breve{Z}$ | $q, X[A\cdot\cdot] \xrightarrow{\varepsilon/w_1 \otimes w_2} r, Y[S]Z[\cdot\cdot]$ | (right pop) |
| $A \xrightarrow{w_1} BC$ | | $q, X[A\cdot\cdot] \xrightarrow{\varepsilon/w_1} q, X[BC\cdot\cdot]$ | (push) |

(b) CFG $\triangleright$ PDA.

| $\mathbf{P}_1$ | $\mathbf{P}_2$ | $\mathbf{P}_1 \triangleright \mathbf{P}_2$ | name |
|---|---|---|---|
| $q_\iota^1, S \xrightarrow{\ell/w_1} q_f^1, \varepsilon$ | $\ell: q_\iota^2, S \xrightarrow{\varepsilon/w_2} q_f^2, \varepsilon$ | $q_\iota^2, S[q_\iota^1, S] \xrightarrow{\varepsilon/w_1 \otimes w_2} q_f^2, \varepsilon$ | (epsilon) |
| $p, A \xrightarrow{\ell/w_1} q_f^1, \varepsilon$ | $\ell: r, X \xrightarrow{a/w_2} s, \varepsilon$ | $r, X[p, A] \xrightarrow{a/w_1 \otimes w_2} s, \varepsilon$ | (terminal) |
| $p, A \xrightarrow{\ell/w_1} q, \varepsilon$ | $\ell: r, X \xrightarrow{\varepsilon/w_2} s, \breve{Y} Z$ | $r, X[p, A\cdot\cdot] \xrightarrow{\varepsilon/w_1 \otimes w_2} s, Y[q, \cdot\cdot]Z[q_\iota^1, S]$ | (left pop) |
| $p, A \xrightarrow{\ell/w_1} q, \varepsilon$ | $\ell: r, X \xrightarrow{\varepsilon/w_2} s, Y \breve{Z}$ | $r, X[p, A\cdot\cdot] \xrightarrow{\varepsilon/w_1 \otimes w_2} s, Y[q_\iota^1, S]Z[q, \cdot\cdot]$ | (right pop) |
| $p, A \xrightarrow{\varepsilon/w_1} q, BC$ | | $r, X[p, A\cdot\cdot] \xrightarrow{\varepsilon/w_1} r, X[q, BC\cdot\cdot]$ | (push) |

(c) PDA $\triangleright$ PDA. States $q_\iota^1$ and $q_\iota^2$ are the initial states of $\mathbf{P}_1$ and $\mathbf{P}_2$, respectively, and similarly for $q_f^1$ and $q_f^2$.

Figure 6: Normal forms of PDA $\triangleright$ CFG, CFG $\triangleright$ PDA, PDA $\triangleright$ PDA resulting from normal forms of their controllers and controllees.

A ***pop computation*** of type $\underset{i}{\overset{X[p,\,A]}{\triangle}}_{j}$ is a partial derivation $X[p, A] \overset{*}{\Rightarrow} s_{(i:j)}$.

**Definition 31.** *Let* $\mathbf{G}$ *be a controller WCFG with nonterminals* $\mathcal{N}$, *and let* $\mathbf{P}$ *be a WLD-PDA with states* $Q$ *and stack alphabet* $\Gamma$. *Let* $s \in \Sigma^*$ *be an input string of length* $|s| = n$. *A* ***pop computation*** *of* $\mathbf{G} \triangleright \mathbf{P}$ *of type* $\underset{i,\,p}{\overset{X[A\,\cdot\cdot]}{\underset{j,\,q\quad k,\,r}{\triangle\,Y[\cdot\cdot]}}}_{l,\,s}$, *where* $0 \le i \le j \le k \le l \le n$, $p, q, r, s \in Q$, $X, Y \in \Gamma$, *and* $A \in \mathcal{N}$, *is a pair of runs* $(\pi_1, \pi_2)$, *where*

$$\pi_1 \in ((p, X[A]) \overset{*}{\Rightarrow} (q, Y[\varepsilon]\mathbf{\Upsilon}_2)) \text{ and scans } s_{(i:j)}$$

$$\pi_2 \in ((r, \mathbf{\Upsilon}_2) \overset{*}{\Rightarrow} (s, \varepsilon)) \text{ and scans } s_{(k:l)}.$$

A ***pop computation*** of type $\underset{i,\,p}{\overset{X[A]}{\triangle}}_{j,\,q}$ is a run $(p, X[A]) \overset{*}{\Rightarrow} (q, \varepsilon)$ scanning $s_{(i:j)}$.

**Definition 32.** *Let* $\mathbf{P}_1$ *be a controller WPDA and* $\mathbf{P}_2$ *be a WLD-PDA, with states and stack alphabets* $Q_1$ *and* $\Gamma_1$, *and* $Q_2$ *and* $\Gamma_2$, *respectively. Let* $s \in \Sigma^*$ *be an input string of length* $|s| = n$. *A* ***pop computation*** *of* $\mathbf{P}_1 \triangleright \mathbf{P}_2$ *of type* $\underset{i,\,p}{\overset{X[e,\,A\,\cdot\cdot]}{\underset{j,\,q\quad k,\,r}{\triangle\,Y[f,\,\cdot\cdot]}}}_{l,\,s}$, *where* $0 \le i \le j \le k \le l \le n$, $p, q, r, s \in Q_2$, $e, f \in Q_1$, $X, Y \in \Gamma_2$, *and* $A \in \Gamma_1$, *is a pair of runs* $(\pi_1, \pi_2)$, *where*

$$\pi_1 \in ((p, X[e, A]) \overset{*}{\Rightarrow} (q, Y[f, \varepsilon]\mathbf{\Upsilon}_2)) \text{ and scans } s_{(i:j)}$$

$$\pi_2 \in ((r, \mathbf{\Upsilon}_2) \overset{*}{\Rightarrow} (s, \varepsilon)) \text{ and scans } s_{(k:l)}.$$

A ***pop computation*** of type $\underset{i,\,p}{\overset{X[e,\,A]}{\triangle}}_{j,\,q}$ is a run $(p, X[e, A]) \overset{*}{\Rightarrow} (q, \varepsilon)$ scanning $s_{(i:j)}$.

## C.2 Deduction Systems

Figures 7 to 9, in the columns labeled "stringsum," show deduction systems for computing stringsums of PDA ▷ CFG, CFG ▷ PDA, and PDA ▷ PDA. The controller and controllee PDAs are assumed to be single-state.

The goal items for PDA ▷ CFG, CFG ▷ PDA, and PDA ▷ PDA are, in order, $\underset{0}{\overset{S[q_\iota,\,S]}{\triangle}}_{n}$, $\underset{0,\,q_\iota}{\overset{S[S]}{\triangle}}_{n,\,q_f}$, and $\underset{0,\,q_\iota^2}{\overset{S[q_\iota^1,\,S]}{\triangle}}_{n,\,q_f}$, where $n$ is the length of the input string.

## D Computing Allsums

**Definition 33.** *Let* $\mathbf{P}$ *be a controller WPDA with states* $Q$ *and stack alphabet* $\Gamma$, *and let* $\mathbf{G}$ *be a WLD-CFG with nonterminal alphabet* $\mathcal{N}$. *A* ***pop computation*** *of* $\mathbf{P} \triangleright \mathbf{G}$ *of type* $\overset{X[p,\,A\,\cdot\cdot]}{\triangle\,Y[q,\,\cdot\cdot]}$, *where* $X, Y \in \mathcal{N}$, $p, q \in Q$, *and* $A \in \Gamma$, *is a pair of partial derivations* $(d_1, d_2)$, *where*

$$d_1 \in (X[p, A] \overset{*}{\Rightarrow} s_1 Y[q, \varepsilon]\mathbf{\Upsilon}_2)$$

$$d_2 \in (\mathbf{\Upsilon}_2 \overset{*}{\Rightarrow} s_2)$$

*and* $s_1, s_2 \in \Sigma^*$, $\mathbf{\Upsilon}_2 \in \mathcal{N}[\Gamma^*]^*$. *The weight of this pop computation is* $\mathbf{w}(d_1, d_2) = \mathbf{w}(d_1) \otimes \mathbf{w}(d_2)$.

A ***pop computation*** of type $\overset{X[p,\,A]}{\triangle}$ is a partial derivation $X[p, A] \overset{*}{\Rightarrow} s$, where $s \in \Sigma^*$.

Figure 7: Deductive systems for computing stringsums and allsums of PDA $\triangleright$ CFG. State $q_\iota$ is the initial state of the controller.

| name | stringsum | allsum | side condition |
| --- | --- | --- | --- |

**(epsilon)**

$$\frac{}{\underset{0,q_\iota \quad n,q_f}{S[S]}} \quad n=0 \qquad \frac{}{\underset{q_\iota \quad q_f}{S[S]}} \qquad q_\iota, S[S] \xrightarrow{\varepsilon/w} q_f, \varepsilon$$

**(terminal)**

$$\frac{}{\underset{i-1,p \quad i,q}{X[A]}} \quad \boldsymbol{s}_i = a \qquad \frac{}{\underset{p \quad q}{X[A]}} \qquad p, S[A] \xrightarrow{a/w} q, \varepsilon$$

**(left pop)**

$$\frac{\underset{j,r \quad k,s}{Z[S]}}{\underset{i,p \quad \underset{i,q \quad j,r}{Y[\cdot\cdot]} \quad k,s}{X[A\,\cdot\cdot]}} \qquad \frac{\underset{r \quad s}{Z[S]}}{\underset{p \quad \underset{q \quad r}{Y[\cdot\cdot]} \quad s}{X[A\,\cdot\cdot]}} \qquad p, X[A\,\cdot\cdot] \xrightarrow{\varepsilon/w} q, Y[\cdot\cdot]Z[S]$$

**(right pop)**

$$\frac{\underset{i,q \quad j,r}{Y[S]}}{\underset{i,p \quad \underset{i,q \quad j,r}{Z[\cdot\cdot]} \quad k,s}{X[A\,\cdot\cdot]}} \qquad \frac{\underset{q \quad r}{Y[S]}}{\underset{p \quad \underset{q \quad r}{Z[\cdot\cdot]} \quad s}{X[A\,\cdot\cdot]}} \qquad p, X[A\,\cdot\cdot] \xrightarrow{\varepsilon/w} q, Y[S]Z[\cdot\cdot]$$

**(push-1)**

$$\frac{\underset{i,q \;\; \underset{j,r \; m,u}{Y[\cdot\cdot]} \;\; o,v}{X[B\,\cdot\cdot]} \qquad \underset{j,r \;\; \underset{k,s \; l,t}{Z[\cdot\cdot]} \;\; m,u}{Y[C\,\cdot\cdot]}}{\underset{i,p \;\; \underset{k,s \; l,t}{Z[\cdot\cdot]} \;\; o,v}{X[A\,\cdot\cdot]}} \qquad \frac{\underset{q \;\; \underset{r \; u}{Y[\cdot\cdot]} \;\; v}{X[B\,\cdot\cdot]} \qquad \underset{r \;\; \underset{s \; t}{Z[\cdot\cdot]} \;\; u}{Y[C\,\cdot\cdot]}}{\underset{p \;\; \underset{s \; t}{Z[\cdot\cdot]} \;\; v}{X[A\,\cdot\cdot]}} \qquad p, X[A\,\cdot\cdot] \xrightarrow{\varepsilon/w} q, X[BC\,\cdot\cdot]$$

**(push-2)**

$$\frac{\underset{i,q \;\; \underset{j,r \; k,s}{Y[\cdot\cdot]} \;\; l,t}{X[B\,\cdot\cdot]} \qquad \underset{j,r \quad k,s}{Y[C]}}{\underset{i,p \quad l,t}{X[A]}} \qquad \frac{\underset{q \;\; \underset{r \; s}{Y[\cdot\cdot]} \;\; t}{X[B\,\cdot\cdot]} \qquad \underset{r \quad s}{Y[C]}}{\underset{p \quad t}{X[A]}} \qquad p, X[A\,\cdot\cdot] \xrightarrow{\varepsilon/w} q, X[BC\,\cdot\cdot]$$

Figure 8: Deductive systems for computing stringsums and allsums of CFG $\triangleright$ PDA. States $q_\iota$ and $q_f$ are the controllee's initial and final states, respectively.

| name | stringsum | allsum | side condition |
|---|---|---|---|
| (epsilon) | $n = 0$ | | $q_\iota^2, S[q_\iota^1, S] \xrightarrow{\varepsilon/w} q_f, \varepsilon$ |
| (terminal) | $s_i = a$ | | $p, X[e, A] \xrightarrow{a/w} q, \varepsilon$ |
| (left pop) | | | $p, X[e, A \cdot\cdot] \xrightarrow{\varepsilon/w} q, Y[f, \cdot\cdot]Z[q, S]$ |
| (right pop) | | | $p, X[e, A \cdot\cdot] \xrightarrow{\varepsilon/w} q, Y[f, S]Z[q, \cdot\cdot]$ |
| (push-1) | | | $p, X[e, A \cdot\cdot] \xrightarrow{\varepsilon/w} q, X[f, BC \cdot\cdot]$ |
| (push-2) | | | $p, X[e, A \cdot\cdot] \xrightarrow{\varepsilon/w} q, X[f, BC \cdot\cdot]$ |

Figure 9: Deductive systems for computing stringsums and allsums of PDA ▷ PDA. States $q_\iota^1$ and $q_\iota^2$ are the initial states of $\mathbf{P}_1$ and $\mathbf{P}_2$, respectively. State $q_f$ is the final state of $\mathbf{P}_2$.

**Definition 34.** *Let* $\mathbf{G}$ *be a controller CFG with nonterminal alphabet* $\mathcal{N}$*, and let* $\mathbf{P}$ *be a WLD-PDA with states* $Q$ *and stack alphabet* $\Gamma$*. A **pop computation** of* $\mathbf{G} \triangleright \mathbf{P}$ *of type* 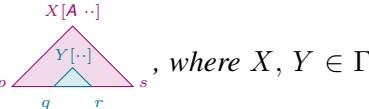*, where* $X, Y \in \Gamma$*, $p, q, r, s \in Q$, and* $A \in \mathcal{N}$*, is a pair of runs* $(\boldsymbol{\pi}_1, \boldsymbol{\pi}_2)$*, where*

$$\boldsymbol{\pi}_1 \in ((p, X[A]) \overset{*}{\Rightarrow} (q, Y[\varepsilon]\boldsymbol{\Upsilon}_2)) \text{ and scans } \boldsymbol{s}_1$$

$$\boldsymbol{\pi}_2 \in ((r, \boldsymbol{\Upsilon}_2) \overset{*}{\Rightarrow} (s, \varepsilon)) \text{ and scans } \boldsymbol{s}_2$$

*and* $\boldsymbol{s}_1, \boldsymbol{s}_2 \in \Sigma^*$*,* $\boldsymbol{\Upsilon}_2 \in \Gamma[\mathcal{N}^*]^*$*. The weight of this pop computation is* $\mathbf{w}(\boldsymbol{\pi}_1, \boldsymbol{\pi}_2) = \mathbf{w}(\boldsymbol{\pi}_1) \otimes \mathbf{w}(\boldsymbol{\pi}_2)$*.*
  *A **pop computation** of type* $\overset{X[A]}{\underset{p \quad q}{\triangle}}$ *is a run* $(p, X[A]) \overset{*}{\Rightarrow} (q, \varepsilon)$ *scanning some* $\boldsymbol{s} \in \Sigma^*$*.*

**Definition 35.** *Let* $\mathbf{P}_1$ *be a controller WPDA and* $\mathbf{P}_2$ *a WLD-PDA. Additionally, let* $Q_1$ *and* $\Gamma_1$*, and* $Q_2$ *and* $\Gamma_2$ *be the states and stack alphabets of* $\mathbf{P}_1$ *and* $\mathbf{P}_2$*, respectively. A **pop computation** of* $\mathbf{P}_1 \triangleright \mathbf{P}_2$ *of type* $\overset{X[e, A\,\cdots]}{\underset{p \quad\; \underset{q\;\;\; r}{Y[f,\cdots]}\;\; s}{\triangle}}$*, where* $p, q, r, s \in Q_2$*, $e, f \in Q_1$, $X, Y \in \Gamma_2$ and* $A \in \Gamma_1$*, is is a pair of runs* $(\boldsymbol{\pi}_1, \boldsymbol{\pi}_2)$*, where*

$$\boldsymbol{\pi}_1 \in ((p, X[e, A]) \overset{*}{\Rightarrow} (q, Y[f, \varepsilon]\boldsymbol{\Upsilon}_2)) \text{ and scans } \boldsymbol{s}_1$$

$$\boldsymbol{\pi}_2 \in ((r, \boldsymbol{\Upsilon}_2) \overset{*}{\Rightarrow} (s, \varepsilon)) \text{ and scans } \boldsymbol{s}_2$$

*and* $\boldsymbol{s}_1, \boldsymbol{s}_2 \in \Sigma^*$*,* $\boldsymbol{\Upsilon}_2 \in \Gamma_2[\Gamma_1^*]^*$*. The weight of this pop computation is* $\mathbf{w}(\boldsymbol{\pi}_1, \boldsymbol{\pi}_2) = \mathbf{w}(\boldsymbol{\pi}_1) \otimes \mathbf{w}(\boldsymbol{\pi}_2)$*.*
  *A **pop computation** of type* $\overset{X[e, A]}{\underset{p \quad q}{\triangle}}$ *is a run* $(p, X[e, A]) \overset{*}{\Rightarrow} (q, \varepsilon)$ *scanning some* $\boldsymbol{s} \in \Sigma^*$*.*

Figures 7 to 9, in the columns labeled "allsum", show deductive systems for computing allsums of PDA ▷ CFG, CFG ▷ PDA, and PDA ▷ PDA. As in §5, the value of the goal item can be computed by fixed-point iteration or the semiring generalization of Newton's method.

## E    Complexity Analysis of Conversions

Our stringsum and allsum algorithms, although designed for the two-level formalisms, can also be used for TAG, LIG, PAA, and EPDA. However, we must apply a series of conversions in order to do this. For instance, we must apply the following transformations to a WLIG with sets of nonterminals $\mathcal{N}$ and stack symbols $\Gamma$ before feeding it into the stringsum algorithm:

1. The controller WPDA and the controllee WCFG have to be extracted from the WLIG, resulting in a WCFG with $\mathcal{O}(|\mathcal{N}|)$ nonterminals and a WPDA with $\mathcal{O}(|\Gamma|)$ stack symbols.

2. The controllee WCFG must be converted into Chomsky normal form. If the productions have at most $k$ symbols on the right-hand side, then the output WCFG has $\mathcal{O}(k \times |\mathcal{R}|)$ nonterminals. Similarly, the controller WPDA must be converted into the normal form. If the WPDA has transitions that push at most $k$ symbols, the resulting WPDA has $\mathcal{O}(k \times |\delta|)$ states.

3. The rules of the controller and of the controllee need to be merged back into a set of WLIG rules. This construction outputs a WLIG with $\mathcal{O}(|\mathcal{N}| \times |Q|)$ nonterminals and $\mathcal{O}(|\Gamma|)$ stack symbols, where $\mathcal{N}$ is the set of nonterminals of the input controllee, and $Q$ and $\Gamma$ are the sets of states and stack symbols of the input controller.