# OpenReview forum: "Efficient Algorithms for Recognizing Weighted Tree-Adjoining Languages"
_EMNLP/2023/Conference — EMNLP 2023 Main_

### Official Review · Reviewer_VNxE · 2023-07-31

**Typos Grammar Style And Presentation Improvements:** on line 447, eficiently -> efficiently
**Soundness:** 4

**Excitement:**

4: Strong: This paper deepens the understanding of some phenomenon or lowers the barriers to an existing research direction.

**Missing References:**

none

**Paper Topic And Main Contributions:**

This paper proposes a new algorithm to compute the probability of a string defined as the sum of the scores of all derivations of an input string and the total score of all derivations in the language for tree adjoining formalisms. The algorithm is based on a two-level representation including a controller and a controllee. Both can be either a context-free grammar (CFG) or a push-down automata (PDA).

Main definitions necessary to understand the paper are included in the preliminaries section. The two level mechanism is then formally defined. On the basis of such definition, an extension of the Chomsky normal form is applied as a basis for following computations. The core of the paper consists on the computation of string sum and all sum, which is based on pop computations defined before. Correctness and complexity analysis are eventually proved. Strength and limitations of the approach are discussed.

My main concern was whether the venue is suitable for this work, which is completely theoretical, without any empirical part in it, nor any discussion or example about potential applications. After rebuttal, I think that if the authors add a discussion about potential applications, the problem can be solved.

**Questions For The Authors:**

none

**Reasons To Accept:**

-- the quantities for which the computation is given are important for the application of the considered mechanisms

-- the two level formalization gives an effective way of representing the formalism

-- the proves seem correct to me

**Reasons To Reject:**

None

**Reproducibility:**

N/A: Doesn't apply, since the paper does not include empirical results.

**Reviewer Confidence:**

3: Pretty sure, but there's a chance I missed something. Although I have a good feel for this area in general, I did not carefully check the paper's details, e.g., the math, experimental design, or novelty.

---

> ### Author Rebuttal · Authors · 2023-08-28
>
> Thank you very much for your review! We greatly appreciate the time and effort put forth in order to understand our paper in detail and review it.
>
>
> ### My main concern is whether the venue is suitable for this work, which is completely theoretical, without any empirical part in it, nor any discussion or example about potential applications.
>
> You are correct that most papers published at *ACL conferences are not purely theoretical. However, we feel that research in formal language theory is still relevant to natural language. For instance, there have been many papers published at *ACL conferences that use similar formalisms for analyzing certain linguistic phenomena (e.g., cross-serial dependencies). Our paper focuses only on speeding up inference in these formalisms but it does not affect any of the results presented in these papers, therefore there is no need to revisit the empirical component. We agree that we should include some discussion about potential applications and we would be happy to do so in the revised version.

---

### Official Review · Reviewer_1qtB · 2023-08-02

**Soundness:** 5

**Excitement:**

4: Strong: This paper deepens the understanding of some phenomenon or lowers the barriers to an existing research direction.

**Paper Topic And Main Contributions:**

This paper builds on the new equivalencies between tree-adjoining grammars, linear indexed grammars, embedded pushdown automata, and pushdown adjoining automata recently proved in Butoi et al 2023. The authors exploit these equivalencies to construct a unified normal form for all of these formalisms, and to introduce semiring-weighted variants using the same "control" mechanism as Butoi et al. They use the unified normal form to develop a novel parsing algorithm which is suitable for the semiring-weighted versions of all four formalisms, and which improves on existing algorithms in terms of time complexity, space complexity, or both.

**Questions For The Authors:**

A. As you note under Limitations (ll. 642-644), it "add[s] extra complexity" to transform a grammar into the equivalent two-level format and then convert it to the proposed normal form. How do these transformations impact grammar size in the worst case, and do you have results showing the actual impact on grammars that have been used in prior empirical work? Does the increased complexity ever outweigh the efficiency gains from your proposed parsing algorithm? (E.g. ll. 555-557 suggest that your approach somehow avoids the added factor of $|\Gamma|$ in the space complexity, but is this not simply offloaded to the initial conversion so that you start from a larger grammar?)

**Reasons To Accept:**

This work describes a unified normal form for all of the common grammar formalisms that generate the class of tree-adjoining languages. Normal forms simplify the task of reasoning about a formalism, and having a unified normal form now trivializes the task of generalizing proofs from one of these formalisms to the others. A well-constructed normal form further simplifies the construction of efficient algorithms, as the authors demonstrate by developing a unified approach to parsing all of these formalisms. This parsing technique achieves state-of-the-art time-complexity, space-complexity, or both depending on the specific formalism under consideration. The authors also generalize the characterization of tree-adjoining formalisms in Butoi et al 2023 to include semiring-weighted variants, which are more practically applicable to NLP contexts where we are often interested in the likelihood of a given grammar producing some output.

**Reasons To Reject:**

I do not see any compelling reasons to reject.

**Reproducibility:**

N/A: Doesn't apply, since the paper does not include empirical results.

**Reviewer Confidence:**

4: Quite sure. I tried to check the important points carefully. It's unlikely, though conceivable, that I missed something that should affect my ratings.

**Typos Grammar Style And Presentation Improvements:**

Consider adding parentheses when writing out PDA transitions, e.g. $(p, X) \xrightarrow{a/w} (q, \gamma)$ on l. 169. Without parentheses, it reads at a glance like the transition is from X to q rather than from a tuple to another tuple.

Ll. 255-256 seem to use different indexing in the equation vs the adjacent text: the equation suggests that $p_1$ is the first production but the text suggests that it should be $p_0$.

Building on my question A, it would be great to see the algorithm complexity stated in terms of the original grammar size (before conversion to a two-level CFG/PDA format) if possible, to clarify which aspects of the complexity derive from the transformation to the required form and which are from the parsing algorithm itself.

---

> ### Author Rebuttal · Authors · 2023-08-28
>
> Thank you very much for your comments and feedback! We greatly appreciate the time and effort put forth in order to understand our paper in detail and help us improve it.
>
> ### A. As you note under Limitations (ll. 642-644), it "add[s] extra complexity" to transform a grammar into the equivalent two-level format and then convert it to the proposed normal form. How do these transformations impact grammar size in the worst case, and do you have results showing the actual impact on grammars that have been used in prior empirical work? Does the increased complexity ever outweigh the efficiency gains from your proposed parsing algorithm? (E.g. ll. 555-557 suggest that your approach somehow avoids the added factor of |Γ| in the space complexity, but is this not simply offloaded to the initial conversion so that you start from a larger grammar?)
>
> We agree that we should discuss the extra complexity added by the conversions from TAG, LIG, EPDA, PAA into their corresponding 2-level formalisms and their conversions into their normal forms, and a comparison between the complexity of our algorithms (including the extra complexity added by these transformations) and the runtime of state-of-the-art algorithms. For instance, a full analysis of the complexity of our algorithm for a LIG, assuming that it has $|\mathcal{N}|$ nonterminals and $|\Gamma|$ stack symbols, would be:
>
> 1. The controller PDA and the controllee CFG can be easily extracted in linear time from the rules of the LIG, resulting in a CFG with  $|\mathcal{N}|$ nonterminals and a single-state PDA with $|\Gamma|$ stack symbols.
> 2. The controllee CFG needs to be converted into Chomsky normal form, and the controller PDA needs to be converted into the normal form. If the PDA has transitions that push at most k symbols, the resulting PDA has $\mathcal{O}(k \times |\delta| \times |Q|)$ states and $\mathcal{O}(k \times |\delta|)$ transitions.
> 3. Generally, the PDA conversion into the normal form will produce a PDA with multiple states, which needs to be converted back into a single-state PDA. If the PDA has $|Q|$ states and $|\Gamma|$ stack symbols, the equivalent single-state PDA will have $\mathcal{O}(|Q|^2 |\Gamma|)$ stack symbols.
> The rules of the controller and of the controllee can be merged back in linear time.
>
> However, the algorithm of Vijay-Shanker and Weir assumes an input LIG in a different normal form, which would be equivalent to a CFG in Chomsky normal form controlled by a simple PDA (Definition 9). So the conversion of a general LIG into such a form would also require similar transformations to ours, increasing the size of the original grammar. We would be happy to use some of the additional page to include such an analysis in the final version of the paper.
>
> ### Consider adding parentheses when writing out PDA transitions, e.g. (p,X)→a/w(q,γ) on l. 169. Without parentheses, it reads at a glance like the transition is from X to q rather than from a tuple to another tuple.
>
> You suggested adding brackets on both sides of a PDA transition, i.e., $(p, X) \xrightarrow{a/w} (q, \gamma)$. However, this might read as a move from a configuration to another configuration, which is inconsistent with our current definition of PDA configuration (a pair consisting of a state and the contents of the stack) since we only include the top of the stack in a transition. We will add brackets around the transitions instead so that they are separated better from the text surrounding them.
>
> ### Ll. 255-256 seem to use different indexing in the equation vs the adjacent text: the equation suggests that p1 is the first production but the text suggests that it should be p0.
>
> In Definition 12, lines 255-256, there is indeed an inconsistency in indexing. The first production of the derivation should be $p_1$ in line 255 and in line 256 there should be $\alpha_{i-1} \xRightarrow{p_i} \alpha_i$ instead of $\alpha_{i-1} \xRightarrow{p_{i-1}} \alpha_i$.

---

### Official Review · Reviewer_F4Ew · 2023-08-04

**Typos Grammar Style And Presentation Improvements:** 097
**Soundness:** 3

**Excitement:**

3: Ambivalent: It has merits (e.g., it reports state-of-the-art results, the idea is nice), but there are key weaknesses (e.g., it describes incremental work), and it can significantly benefit from another round of revision. However, I won't object to accepting it if my co-reviewers champion it.

**Missing References:**

Pop computations have been discussed a lot in the literature on dynamic programming techniques to simulate PDAs. I think you should add some references in Section 5.

698: bib reference is incomplete, add name of the book.

**Paper Topic And Main Contributions:**

This draft considers the class of languages generated/recognized by tree-adjoining grammars or, equivalently, linear indexed grammars, pushdown-adjoining automata, and embedded pushdown automata. It is known that all of these formalisms can be viewed as two-level formalisms, where some controller formalism is used to control the derivations/computations of a second formalism, here called the controllee formalism. Using the combination of standard normal forms for both controller and controllee formalisms, and using dynamic programming techniques, the draft develops novel parsing algorithms for the weighted languages generated by tree-adjoining grammars.

**Questions For The Authors:**

A.
There is something about the definitions of WLD-CFG and WLD-PDA that confuses me: these two definitions are not fully symmetrical as I was expecting. WLD-CFG has special rules X[A..] -> X[..] that are used to simulate non-scanning transitions of the PDA controller, or else are used to simulate leftmost derivations of the CFG controller when the sentential form starts with a nonterminal. What is the corresponding feature for WLD-PDA?

B.
d-strong equivalence is mentioned in two places in this draft, and seems a very important notion related to your findings. Unfortunately, there is no definition of d-strong equivalence in this draft, as far as I can tell.

C.
There is a technical problem with your definition of pop computation. As an example, consider definition 24 where the controllee is a WLD-CFG and the controller is a WPDA. In derivation d the controller moves from configuration (q, A\gamma) to configuration (q, \gamma). However, it could well be that at the intermediate steps the topmost symbols of \gamma are popped from the stack and later replaced by exactly their same copies (by some push transition). Those computations have type [i,X,A,j,k,Y,l] according to your definition, but are *not* accounted for by your deductive rule system in figure 2. The idea that pop computations should not 'consume' the part of the stack \gamma below A has been extensively discussed in the literature, see for instance Kuhlmann, Gómez-Rodríguez, Satta 2011, Dynamic Programming Algorithms for Transition-Based Dependency Parsers, and references therein. You have a similar problem in the related definitions in section 5.

D.
There is an overloading between the signature [i,X,A,j,k,Y,l] that you use to indicate types for pop computations, and the notion of item [i,X,A,j,k,Y,l] used by your deductive rule system. Of course these objects have related meaning, but I think you need to resolve this conflict.

E.
I think there is a problem with theorem 2. The theorem establishes a correspondence between items and pop computations together with derivations of the form Y[\alpha] =*=> s(k:l]. However, notice that the derivation Y[\alpha] =*=> s(k:l] is not checked by your deductive rule system: this is apparent from rules (22) and (23) where you make a 'guess' on the extension of the gap (i--j in rule 22) without checking for the existence of the associated derivation.

-------
This part written after the Author responses:

A: ok.

B: I am already familiar with the notion of d-strong equivalence, double dot notation for LIGs, etc.  My point is that this draft is not self contained: the average ACL reader will not be able to understand your theoretical developments without a basic discussion of the above notions. Using specialised mathematical notation without providing a few lines of explanations, as those that you wrote in your response, does not meet the standard of scholarship of the EMNLP conference. I recommend you fix this in the final version, if you have extra space.

C: the new definition is ok.

D: ok.

E: you are right, sorry I misread steps (22) and (23).  You should be aware that the implementation of your deductive rule system should ensure that computation of weights w[j,Z,k] should be completed before undergoing steps (22) and (23).

Accordingly, I have changed my overall score from 2 to 3. I wanted to provide 3.5, but this option is not available.

**Reasons To Accept:**

Novel results, interesting combination of existing techniques.

**Reasons To Reject:**

Technical presentation is not carefully developed, some parts of the draft are confusing or mistaken, and need further elaboration.

**Reproducibility:**

5: Could easily reproduce the results.

**Reviewer Confidence:**

4: Quite sure. I tried to check the important points carefully. It's unlikely, though conceivable, that I missed something that should affect my ratings.

---

> ### Author Rebuttal · Authors · 2023-08-28
>
> Thank you very much for your comments and feedback! We greatly appreciate the time and effort put forth in order to understand our paper in detail and help us improve it.
>
> # Responses to "Questions For The Authors"
>
> ### A. There is something about the definitions of WLD-CFG and WLD-PDA that confuses me: these two definitions are not fully symmetrical as I was expecting. WLD-CFG has special rules X[A..] -> X[..] that are used to simulate non-scanning transitions of the PDA controller, or else are used to simulate leftmost derivations of the CFG controller when the sentential form starts with a nonterminal. What is the corresponding feature for WLD-PDA?
>
> We apologize for not explaining this point more clearly.The definitions of WLD-CFG and WLD-PDA are fully symmetrical. You mentioned a rule of the form $X[A \cdot \cdot] \rightarrow X[\cdot \cdot]$, which simulates the application of a production rule/transition of the controller CFG/PDA, without the application of any rule of the controllee.
>
> The WLD-PDA counterparts would be $p, X[A \cdot \cdot] \xrightarrow{\varepsilon} p, X[\cdot \cdot]$, for a CFG controller, and  $p, X[r, A \cdot \cdot] \xrightarrow{\varepsilon} p, X[s, \cdot \cdot]$, for a PDA controller. You can observe this symmetry more clearly in Figure 1, where there is a one-to-one correspondence between the rules of the four formalisms. The types of rules that you mentioned are stated in equations (5), (10), (15), and (20). We will add some extra explanations in the revised version.
>
> ### B. d-strong equivalence is mentioned in two places in this draft, and seems a very important notion related to your findings. Unfortunately, there is no definition of d-strong equivalence in this draft, as far as I can tell.
>
> The definition of d-strong equivalence is indeed crucial for our findings. The definition was originally introduced by Butoi et al. (2023), together with the definitions of the two-level formalisms, which we extended to the weighted case in our paper. We agree that our paper would be improved by providing this definition and we will do so in the final version.
>
> ### Section 3 introduces two-level formalisms without much discussion of the intuition behind the formalisms. I think this makes it very difficult for the reader to follow the developments of your findings. For instance, the intuition behind the tilde notation on rules is not presented. Same thing for the double dot notation that appears for the first time at l.230 and is used later on in your proofs.
>
> You mentioned that Section 3 introduces the two-level formalisms without much intuition behind them. The two-level formalisms are not our contribution, they were introduced in Butoi et al. (2023), which contains a much more detailed presentation, including some informal explanations. Our contribution in this paper is a weighted version of their definitions. The double dot notation which first appears in line 230 is standard in the context of LIGs. The double dots represent an arbitrary sequence of stack symbols. The nonterminal symbols with a breve (we think that’s what you mean by tilde) on top are called distinguished nonterminals and they were first introduced by Weir (1992), and then used also by Butoi et al. (2023). The distinguished nonterminal is the nonterminal on the right-hand side of a controllee production which receives the controller sentential form/configuration after the application of the production.
>
> ### C. There is a technical problem with your definition of pop computation. As an example, consider definition 24 where the controllee is a WLD-CFG and the controller is a WPDA. In derivation d the controller moves from configuration (q, A\gamma) to configuration (q, \gamma). However, it could well be that at the intermediate steps the topmost symbols of \gamma are popped from the stack and later replaced by exactly their same copies (by some push transition). Those computations have type [i,X,A,j,k,Y,l] according to your definition, but are not accounted for by your deductive rule system in figure 2. The idea that pop computations should not 'consume' the part of the stack \gamma below A has been extensively discussed in the literature, see for instance Kuhlmann, Gómez-Rodríguez, Satta 2011, Dynamic Programming Algorithms for Transition-Based Dependency Parsers, and references therein. You have a similar problem in the related definitions in section 5.
>
> You are correct - thank you for spotting this! Indeed, we meant that $\gamma$ from your example should not be touched in any intermediate configurations of the controller. This is reflected in our deduction system, which only accounts for this type of derivation. The correct version of Definition 24 is the following:
>
> Let $\mathbf{F}_1$ be a WLD-CFG, $\mathbf{F}_2$ a controller WPDA with a single state $q$, and $\mathbf{s} \in \Sigma^*$ a string of length $|\mathbf{s}|=n$. A pop computation of $\mathbf{F}_2 \triangleright \mathbf{F}_1$ of type $[i,X,A,j,k,Y,l]$, where $0 \leq i \leq k \leq l \leq j \leq n$, $X,Y \in \mathcal{X}_1$, and $A \in \mathcal{X}_2$, is a derivation $d = X_0[q, \gamma_0],\Upsilon_1 X_1[q, \gamma_1] \Upsilon’_1, \ldots, \Upsilon_m X_m[q, \gamma_m] \Upsilon'_m, \ldots, \mathbf{s}(i,j]$ of the substring $\mathbf{s}(i,j]$, where $X_0=X$, $X_m=Y$, $\gamma_0=A \gamma_m$ and for all $l<m$, $|\gamma_l| \ge |\gamma_0|$, such that there is a derivation $d' =  Y[q, \gamma], \ldots, \mathbf{s}(k,l]$ of the substring $\mathbf{s}(k,l]$. A pop computation of type $[i,X,A,j,\bot,\bot,\bot]$ is a derivation $d = X[q, A], \ldots,\mathbf{s}(i,j]$.
>
> ### D. There is an overloading between the signature [i,X,A,j,k,Y,l] that you use to indicate types for pop computations, and the notion of item [i,X,A,j,k,Y,l] used by your deductive rule system. Of course these objects have related meaning, but I think you need to resolve this conflict.
>
> We will consider rewriting the definitions of pop computations in the revised version such that they don’t include the signature [i,X,A,j,k,Y,l] and keeping the items only in the deduction system, similar to Kuhlmann et al. (2011).
>
> ### E. I think there is a problem with theorem 2. The theorem establishes a correspondence between items and pop computations together with derivations of the form Y[\alpha] ==> s(k:l]. However, notice that the derivation Y[\alpha] ==> s(k:l] is not checked by your deductive rule system: this is apparent from rules (22) and (23) where you make a 'guess' on the extension of the gap (i--j in rule 22) without checking for the existence of the associated derivation.
>
> The deduction system computes weights of derivations with a “gap”, where the weight of the gap is not accounted for (otherwise we would overcount some parts of the derivations when deriving the weights of other items). The derivation Y[\alpha] ==> s(k:l] is not checked by our deduction system because this derivation represents the gap of a pop computation. But, if such a derivation exists, its weight will be used to compute the weight of the full derivation, including the gap.
>
> # Response to "Typos, Grammar Style, And Presentation Improvements"
>
> ### Proof of theorem 1: I don't really see the need for this proof, why don't you just say something like 'directly follows from the definition of item'?
>
> Regarding the proof of Theorem 1, we find it necessary for proving the correctness of the deduction system. While it is true that we provide an informal description of the items of the form $[i,X,j]$ and their weights in lines 457--461, we currently don’t have any other formal definition for this type of items (we currently only have definitions for pop computations and their corresponding items), therefore the theorem formalizes this intuition. The proof also does not immediately follow from the definition, as it requires the application of rule from equation (25) (although, we do agree that this step is straightforward).
>
> ### 184: your definition 9 is not used anywhere else in the draft, as far as I can see ... I wonder why you want to have a numbered definition for some notion that is never used.
>
> You mentioned that the simple PDA introduced in Definition 9 is not used anywhere in the paper. The type of LIG used by Vijay–Shanker and Weir in their parsing algorithm is equivalent to an LD-CFG controlled by a simple PDA. The simple PDA does not allow computing stringsums efficiently (this is explained in more detail in Butoi et al. (2022)), therefore two-level formalisms having such controller PDAs have the same issue. We agree that we should explain this point in greater detail and we would be happy to do so in the final version.
>
>
> We also thank you for spotting the typos (e.g., the ones in lines 150, 164, 302, etc.) and the missing references (e.g., Kuhlmann et al., 2011, Butoi et al., 2022 for pop computations) in the current version of the paper and for your suggestions (e.g., for lines 456-457).

---

### Meta-Review · Area_Chair_hpTa · 2023-09-18

**Recommendation:** 5

**Metareview:**

This paper proposes a unified normal form for the class of tree-adjoining languages, the two-level representation leads to efficient algorithms to parse all those formalisms. Typically, they introduce new definitions of semiring-weighted two-level formalisms, propose new stringsum  and allsum algorithms, and prove the correctness and efficiency of their algorithms. Those formalisms and findings are beneficial to the community of tree-adjoining languages, and may lead to some potential applications.

---

### Decision · Program_Chairs · 2023-10-07

**Decision:**

Accept-Main

**Comment:**

This paper proposes a unified normal form for the class of tree-adjoining languages, the two-level representation leads to efficient algorithms to parse all those formalisms. Typically, they introduce new definitions of semiring-weighted two-level formalisms, propose new stringsum  and allsum algorithms, and prove the correctness and efficiency of their algorithms. Those formalisms and findings are beneficial to the community of tree-adjoining languages, and may lead to some potential applications.